# S-Trimer, a COVID-19 subunit vaccine candidate, induces protective immunity in nonhuman primates

Joshua G. Liang[1,6], Danmei Su[1,6], Tian-Zhang Song[2,6], Yilan Zeng [3,6], Weijin Huang [4,6], Jinhua Wu[1], Rong Xu[1], Peiwen Luo[1], Xiaofang Yang[1], Xiaodong Zhang[1], Shuangru Luo[1], Ying Liang[1], Xinglin Li[1], Jiaju Huang[1], Qiang Wang[1], Xueqin Huang[1], Qingsong Xu[1], Mei Luo[3], Anliang Huang[5], Dongxia Luo[3], Chenyan Zhao[4], Fan Yang[5], Jian-Bao Han[2], Yong-Tang Zheng [2] & Peng Liang [1✉]

SARS-CoV-2 is the underlying cause for the COVID-19 pandemic. Like most enveloped RNA viruses, SARS-CoV-2 uses a homotrimeric surface antigen to gain entry into host cells. Here we describe S-Trimer, a native-like trimeric subunit vaccine candidate for COVID-19 based on Trimer-Tag technology. Immunization of S-Trimer with either AS03 (oil-in-water emulsion) or CpG 1018 (TLR9 agonist) plus alum adjuvants induced high-level of neutralizing antibodies and Th1-biased cellular immune responses in animal models. Moreover, rhesus macaques immunized with adjuvanted S-Trimer were protected from SARS-CoV-2 challenge compared to vehicle controls, based on clinical observations and reduction of viral loads in lungs. Trimer-Tag may be an important platform technology for scalable production and rapid development of safe and effective subunit vaccines against current and future emerging RNA viruses.

[1] Clover Biopharmaceuticals, Chengdu, China. [2] Key Laboratory of Animal Models and Human Disease Mechanisms of the Chinese Academy of Sciences, Kunming Institute of Zoology, Chinese Academy of Sciences, Kunming, China. [3] Public Health Clinical Center of Chengdu, Chengdu, China. [4] Division of HIV/ AIDS and Sex-Transmitted Virus Vaccines, Institute for Biological Product Control, National Institutes for Food and Drug Control (NIFDC), Beijing, China. [5] Department of Pathology, Chengdu Fifth People's Hospital, Chengdu, China. [6] These authors contributed equally: Joshua G. Liang, Danmei Su, Tian-Zhang Song, Yilan Zeng, Weijin Huang. ✉email: liang.peng@cloverbiopharma.com

Despite tremendous progress made in the past century in eradicating numerous infectious diseases via vaccinations, enveloped RNA viruses remain a major threat to global public health. Even with partially-effective vaccines available, seasonal influenza can lead to hundreds of thousands of deaths worldwide each year[1]. Public health experts have warned that a new lethal viral strain of influenza resembling the 1918 H1N1 pandemic strain, which led to an estimated 50 million deaths worldwide a century ago, may one day emerge again[2]. To the surprise of many, coronaviruses that most-often cause mild common cold have become a major threat to public health since the emergence of SARS-CoV (severe acute respiratory syndrome coronavirus) in 2003[3], followed by MERS-CoV (Middle East respiratory syndrome coronavirus) in 2012[4]. In 2020, COVID-19 caused by the SARS-CoV-2 virus has become a global pandemic with over 30 million people infected and a death toll of nearly one million people as of September 2020[5]. The fact that no effective vaccines have ever been successfully developed against RSV, HIV or any coronavirus, all of which are enveloped RNA viruses, since they were first discovered decades ago, is a reminder of the challenges faced in developing a vaccine for SARS-CoV-2.

Similar to other enveloped RNA viruses (such as RSV, HIV, and influenza), coronaviruses including SARS-CoV-2 also use a distinct trimeric antigen (Spike protein) on their viral envelopes to gain entry into its host cells. The trimeric Spike (S) protein of SARS-CoV-2 binds to ACE2 (angiotensin-converting enzyme 2), the host cell surface receptor, and mediates subsequent viral entry via membrane fusion[6]. COVID-19 symptoms range widely, from mild flu-like symptoms to severe fatal disease that is characterized by pneumonia and often accompanied by multi-organ system failures. Severe disease appears to more often occur in certain populations such as the elderly and individuals with risk-factors such as immune-related diseases, cardiovascular disease, and diabetes[7,8]. It is hoped that a safe and effective vaccine for SARS-CoV-2 can be rapidly developed in order to end this global pandemic. To this end, multiple vaccine strategies have been deployed and have entered into clinical testing, including mRNA and DNA vaccines, adenovirus-based viral vectors, inactivated SARS-CoV-2, and protein-based subunit vaccines[9]. A successful COVID-19 vaccine that could impact the course of the SARS-CoV-2 pandemic must have four essential characteristics: safety, efficacy, scalability of manufacturing and distribution, and speed of development[10].

Challenges for developing safe and effective vaccines for some enveloped RNA viruses are hampered by two major obstacles: difficulties in inducing broadly neutralizing antibodies such as in the case of HIV and risk of vaccine-associated enhanced respiratory disease (VAERD) in the case of RSV[11] and SARS-CoV[12]. Recombinant subunit HIV vaccines using either monomeric or dimeric gp120 antigens failed in multiple clinical studies due to lack of efficacy[13]. More recently, the revelation that asymptomatic HIV-positive carriers can produce broadly neutralizing antibodies that only recognize the native trimeric gp140, but not monomeric gp120[14–17], implies that preserving the native trimeric conformation of viral antigens may be important to the future success of vaccines for HIV as well as other enveloped RNA viruses. The risk of disease enhancement (VAERD) has been a key challenge for the development of vaccines for RNA viruses causing respiratory disease, as one inactivated RSV vaccine caused clear disease enhancement and more frequent hospitalizations than placebo in clinical trials decades ago[11], and similar observations have been made for SARS-CoV vaccine candidates tested in animal models[12]. These challenges and risks may also be encountered in the development of COVID-19 vaccines[9].

Here we describe the use of a platform technology dubbed as Trimer-Tag[18] with a tailored affinity purification scheme to rapidly produce (in mammalian cells) a native-like prefusion form of trimeric Spike protein subunit antigen derived from the wild-type viral sequence as a COVID-19 vaccine candidate. We demonstrate that adjuvanted S-Trimer induced high-levels of neutralizing antibodies as well as Th1-biased cellular immune responses in animal models and protected nonhuman primates from SARS-CoV-2 challenge.

## Results

**High-level expression, purification, and characterization of S-Trimer antigen.** To rapidly express the S-Trimer antigen, we employed Trimer-Tag technology[18]. cDNA encoding the ectodomain of wild-type SARS-CoV-2 Spike (S) protein (Fig. 1a) was subcloned into the pTRIMER mammalian expression vector to allow in-frame fusion to Trimer-Tag, which is capable of self-trimerization via disulfide bonds (Fig. 1b). After stable transfection into CHO cells, subsequent screening for high-titer production clones, and extensive process optimization, a fed-batch serum-free cell culture process in bioreactor was developed leading to high-level expression of S-Trimer as a secreted protein with a titer ~500 mg/L (Fig. 1c).

To obtain S-Trimer in a highly purified form for vaccine studies, we developed an affinity purification scheme (fig. S1a), by taking advantage of the high binding-affinity between Trimer-Tag and Endo180, a collagen receptor capable of binding to the C-terminal region of Type 1 procollagen[19] and to mature collagen[20]. Endo180-Fc fusion protein (fig. S1b) was loaded onto a Protein A column and captured by the resins via high-affinity binding between Protein A and human IgG1 Fc domain of Endo180-Fc. Then, serum-free cell culture medium containing S-Trimer secreted by CHO cells was loaded onto the Protein A column with pre-captured Endo180-Fc. After washing off any unbound contaminating host cell proteins (HCP) and other impurities, the bound S-Trimer was purified to near homogeneity in a single step using moderate salt elution, conditions that do not dissociate Endo180-Fc from the Protein A column (fig. S1c). S-Trimer was further purified through low pH for preventative viral inactivation (VI), anion exchange chromatography to remove host cell DNA and any residual endotoxins, nanofiltration as a preventative viral removal (VR) step and finally UF/DF to concentrate S-Trimer to the desired concentration in formulation buffer to obtain active drug substance (DS) of S-Trimer subunit vaccine candidate (Fig. 1c and fig. S1c). Stability analysis of purified S-Trimer indicates that S-Trimer is stable in liquid solution formulations at 2–8 °C for at least 6 months (fig. S2). Longer-term stability studies are on going.

SDS-PAGE analysis under both non-reducing and reducing conditions confirmed that the purified S-Trimer was a disulfide bond-linked trimer and partially cleaved at S1/S2 boundary by furin protease (Fig. 1d), which is produced by CHO cells[21]. Western blot analysis using either a polyclonal antibody detecting Trimer-Tag or two monoclonal antibodies specific to the S1 and S2 domain, respectively, confirmed the structural features and integrity of the S-Trimer fusion protein (fig. S1d). Under non-reducing conditions, S-Trimer appeared in multiple high molecular weight forms, likely as a result of partial cleavage of the antigen, with non-covalently linked and cleaved S1 released during sample treatment. The reduced form of uncleaved S-Trimer has a molecular weight of around 245 kDa. Peptide sequencing via Edman degradation confirmed S1/S2 cleavage between 685R-686S and also revealed that the N-terminal amino acid of S-Trimer was blocked after signal peptide cleavage between 13S-14Q, likely via pyroglutamate formation at residual 14Q. This was confirmed by subsequent peptide sequencing after pyroglutamate aminopeptidase removal of 14Q (fig. S3). Protein

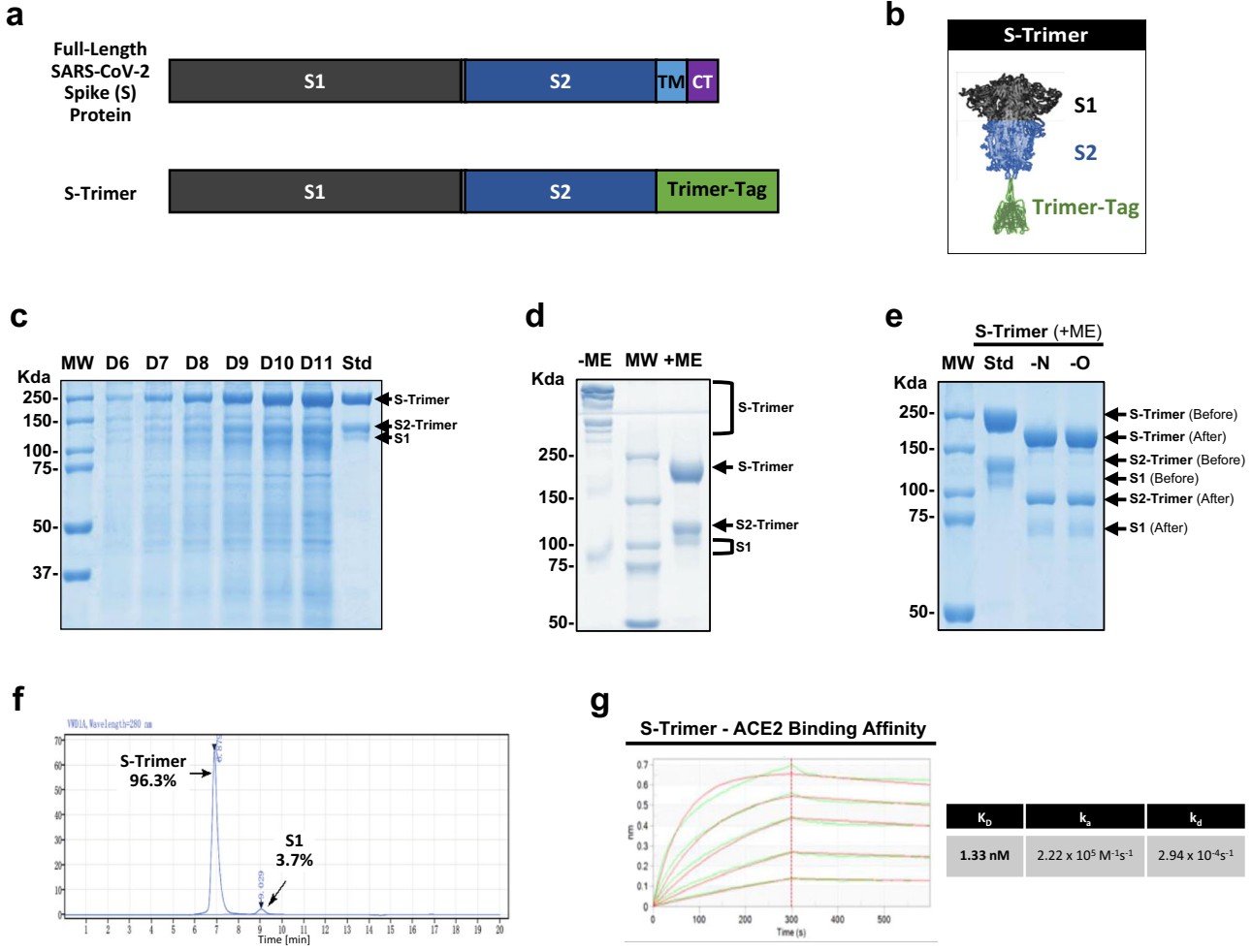

**Fig. 1 High-level expression and characterization of S-Trimer. a** Schematic representations of full-length SARS-CoV-2 Spike (S) protein and the ectodomain of wild-type SARS-CoV-2 S protein-Trimer-Tag fusion protein (S-Trimer). **b** Schematic 2-D illustration of S-Trimer with homotrimeric Spike protein in the prefusion conformation. **c** Reducing SDS-PAGE analysis with Coomassie Blue staining of high-level expression of S-Trimer as a secreted protein from CHO cells in a 15L bioreactor Fed-batch serum-free culture over 11 days (10 μL of cleared media were loaded for each sample) along with a purified standard (Std). **d** S-Trimer is a disulfide bond-linked homotrimer as analyzed by SDS-PAGE with Coomassie Blue staining under non-reducing (-ME) and reducing (+ME) conditions. S-Trimer was shown to be partially cleaved at S1/S2 junction as indicated. **e** S-Trimer is heavily N-glycosylated. Analysis of S-Trimer before and after deglycosylation with PNGase F (-N) and PNGase F & Endo-O (-O) by SDS-PAGE with Coomassie Blue staining under reducing (+ME) condition. **f** SEC-HPLC analysis of the purity of S-Trimer with an MW of approximately 700 Kda, and a small fraction of cleaved S1 was shown detached from S-Trimer as indicated. **g** Determination of the binding affinity between S-Trimer and human ACE2-Fc by ForteBio BioLayer interferometry. All of the above results are representatives of at least three independent experiments.

glycosylation of S-Trimer was analyzed by N- and O-linked deglycosylases, which showed extensive N-linked glycosylation at both S1 and S2 regions, accounting for about 32% mass (79 kDa) to be glycans based on molecular weight changes of S2-Trimer and S1 before (129 kDa and 116 kDa) and after deglycosylation (93 kDa and 72 kDa; Fig. 1e). The purity of purified S-Trimer was analyzed by size-exclusion SEC-HPLC showing a 96.3% main peak ~700 Kda and a 3.7% minor peak ~180 Kda identified as cleaved S1 (Fig. 1f and fig. S3). The binding affinity ($K_D$) of purified S-Trimer to the human ACE2 receptor using ForteBio BioLayer interferometry was shown to be 1.33 nM (Fig. 1g). Negative-stain EM visualization confirmed that S-Trimer particles exist predominantly in a metastable, trimeric prefusion form resembling the full-length wild-type spike protein (fig. S4), which was further confirmed by cryo-EM structural studies[22].

**Detection of SARS-CoV-2-specific binding and neutralizing antibodies in convalescent sera with S-Trimer.** S-Trimer was used as an antigen to detect the presence of SARS-CoV-2 Spike protein binding antibodies and ACE2-competitive antibodies in 41 human convalescent sera samples collected from recovered COVID-19 patients. High levels of S-Trimer binding antibody and ACE2-competitive titers were detected in the convalescent sera, as well as high neutralizing antibody titers using a pseudo-virus neutralization assay (Fig. 2a). S-Trimer binding antibodies were not detected in the sera of naïve human volunteers (fig. S5), whereas antibodies binding to influenza hemagglutinin (HA)-Trimers were detected in both COVID-19 convalescent sera and naïve sera, implying prior infection by influenza in all subjects tested but only SARS-CoV-2 infection in the COVID-19 convalescent subjects. These results support the specificity of the assay and demonstrate the ability of S-Trimer to detect SARS-CoV-2 Spike protein-specific antibodies in convalescent sera, further confirming the native-like conformation of the Spike antigen in S-Trimer.

Analysis of antibody titers detected in the convalescent sera was stratified based on various factors (including COVID-19

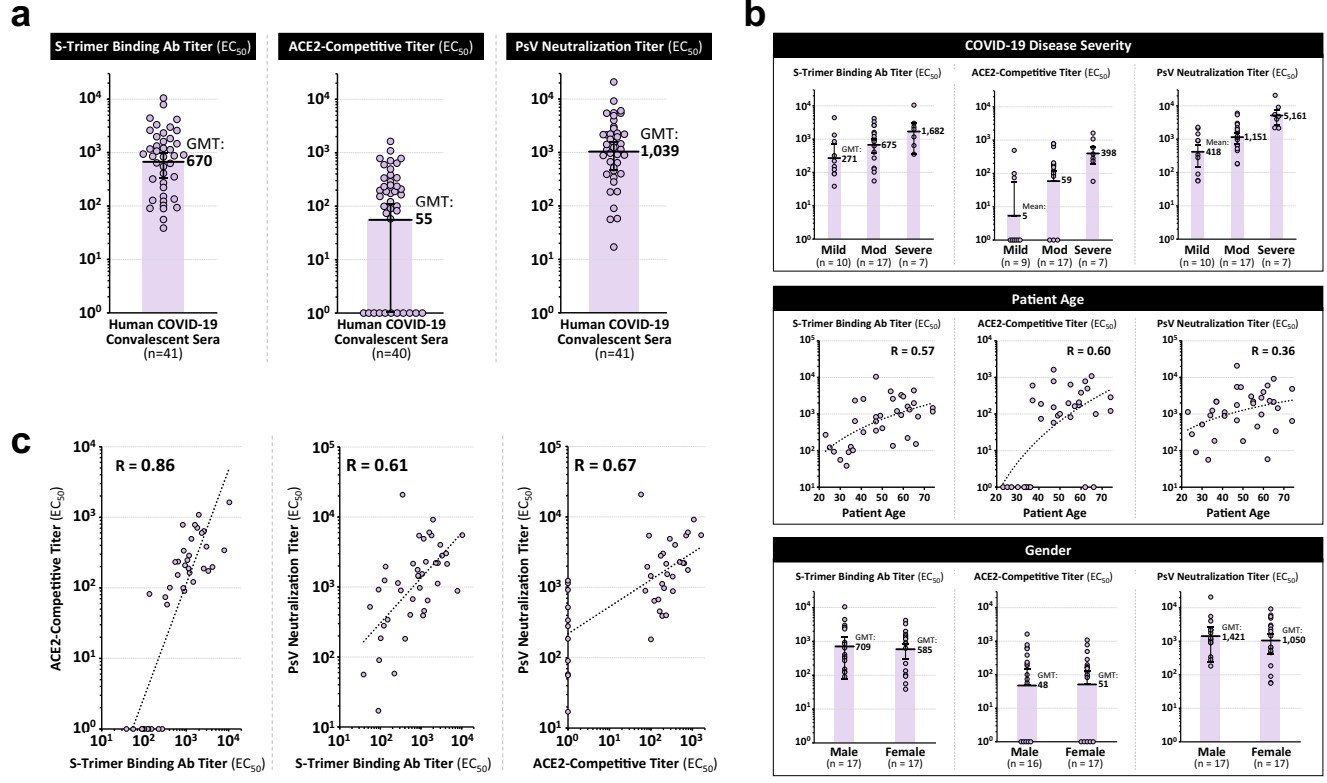

**Fig. 2 Detection of SARS-CoV-2 spike binding and neutralizing antibodies in human COVID-19 convalescent sera.** 41 human convalescent sera collected from recovered COVID-19 subjects were analyzed for **a** S-Trimer binding antibody titers, ACE2-competitive titers, and pseudovirus neutralization titers. **b** Antibody titers were stratified based on patient COVID-19 disease severity, patient age, and gender in 34 subjects where such information was available. **c** Antibody titers in the human convalescent sera for the three assays (S-Trimer binding antibodies, ACE2-competitive, and pseudovirus neutralization) were analyzed for correlation by using two-tailed Pearson's R analysis. Points represent individual humans; horizontal lines indicate geometric mean titers (GMT) of $EC_{50}$ for each group ±SEM.

disease severity, patient age and patient gender) in 34 subjects for whom such information was available (Fig. 2b and table S1). Antibody titers were observed to correlate with disease severity, with lower antibody titers observed in patients with mild COVID-19 disease and higher titers in severe cases, consistent with findings from other published studies[23,24]. ACE2-competitive antibody titers were detectable in only 33% ($n = 3/9$) of patients with mild disease, while these antibodies were present in 82% ($n = 14/17$) of patients with moderate disease and in 100% ($n = 7/7$) in patients with severe disease. Antibody titers also appeared to moderately correlate with patient age, but no differences were observed between genders.

Antibody titers in human convalescent sera were observed to be correlated between the three assays utilized (Fig. 2c), and these correlations were further confirmed in sera from animals immunized with S-Trimer (fig. S6 and table S2). Interestingly, several convalescent sera samples with detectable pseudoviral neutralizing antibody titers did not have any detectable ACE2-competitive titers (Fig. 2c, right), suggesting that either some RBD binding antibodies that do not interfere with ACE2 receptor binding exist, or other domains such as NTD and S2 may also be important antigenic epitopes for viral neutralization as previously reported[25,26].

**Immunogenicity of S-Trimer in rodents.** The immunogenicity of S-Trimer was first evaluated in BALB/c mice. Mice were vaccinated intramuscularly twice in a two-dose prime-boost regimen (Days 0 and 21) with S-Trimer either non-adjuvanted or with various adjuvants including AS03, CpG 1018, and CpG 1018 plus alum. The adjuvant effects on humoral immunogenicity were

evident, as S-Trimer binding antibody titers, ACE2 competitive titers and neutralizing antibody titers in the adjuvanted groups were significantly higher than non-adjuvanted vaccine at corresponding antigen dose levels (Fig. 3a–c). High levels of neutralizing antibody titers were only observed in AS03 and CpG 1018 plus alum-adjuvanted groups (Fig. 3c), but not for non-adjuvanted S-Trimer nor with CpG 1018 alone-adjuvanted S-Trimer. S-Trimer adjuvanted with either AS03 or CpG 1018 plus alum elicited both ACE2-competitve and pseudovirus neutralizing antibody titers similar to or higher than levels observed in human convalescent sera samples. Similar results were observed in rats immunized with S-Trimer (fig. S7), albeit at higher overall antibody titers than in the mouse studies likely in large-part due to the administration of adjuvants (AS03, CpG 1018, and CpG 1018 plus alum) at intended-human dose levels (10-fold higher for AS03 and 75- to 150-fold higher for CpG 1018 compared to doses used in mouse studies). To determine if the S antigen in trimeric form is important in evoking optimal immune response that is crucial for immune protection, we then compare the immunogenicity of S-Trimer and S-Fc (a dimeric form of full-length soluble S protein from SARS-Cov-2) fusion proteins in mice. As expected, after immunization of the animals, S-Trimer with AS03 adjuvant induced a significantly higher neutralizing antibody titers than that of dimeric S antigen with the same adjuvant (fig. S8), despite their binding antibody titers were comparable.

S-Trimer antigen-specific cell-mediated immunity (CMI) was studied by harvesting splenocytes from immunized mice at killing, followed by stimulation with S-Trimer antigen and detection of Th1 (IL-2 and IFNγ) and Th2 (IL-4 and IL-5)

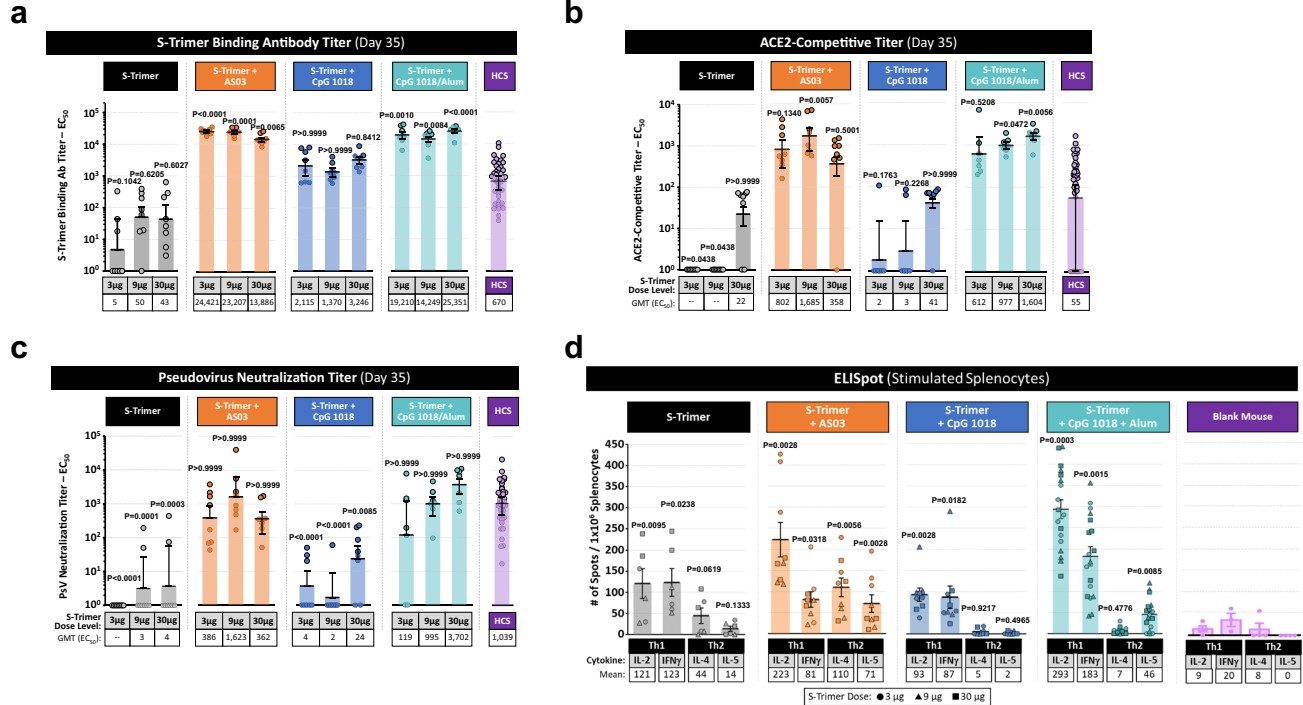

**Fig. 3 Immunogenicity of S-Trimer in mice.** BALB/c mice ($n = 7$–8/group) were immunized with various doses of S-Trimer that was non-adjuvanted or adjuvanted with 25 µL AS03, 10 µg CpG 1018, or 10 µg CpG 1018 plus 50 µg alum twice on Day 0 and Day 21. The humoral immune responses on Day 35 were analyzed and compared with a human convalescent sera (HCS) panel ($n = 41$), based on **a** S-Trimer binding antibody ELISA titers, $n = 7$–8, **b** ACE2-competitive ELISA titers, $n = 7$–8, and **c** SARS-CoV-2 pseudovirus neutralization titers, $n = 7$–8. After necropsy, splenocytes were harvested from mice and stimulated with S-Trimer antigen, followed by **d** detection of Th1 (IL-2, IFNγ) and Th2 (IL-4, IL-5) cytokines by ELISpot. ELISpot data shown represents pooled data across S-Trimer doses (S-Trimer group, $n = 6$; S-Trimer+AS03 group, $n = 9$; S-Trimer+CpG group, $n = 9$; S-Trimer+alum+CpG group, $n = 18$; and Blank mouse group, $n = 3$–4). Points represent individual animals or humans; horizontal lines indicate geometric mean titers (GMT) for antibody assays and mean values for ELISpot assay for each group ±SEM. For statistical analysis of antibody titers, all comparisons were made against HCS sera using Kruskal–Wallis ANOVA with Dunn's multiple comparisons tests. In the ELISpot assays, for all cytokines, the comparisons were compared to blank mouse control with Two-tailed Mann–Whitney tests. P values < 0.05 were considered significant.

cytokines by ELISpot. The CpG 1018 plus alum and AS03 groups appeared to induce a stronger overall CMI response than non-adjuvanted S-Trimer (Fig. 3d). A Th1-biased cell-mediated immune response was observed across non-adjuvanted and CpG 1018-adjuvanted (with or without alum) S-Trimer groups, while a mixed Th1-Th2 profile was observed for AS03. CMI did not appear to be dependent on the dose of antigen used, and it is possible further decrease in the amount of antigen would be necessary to observe the linear range of response in CMI (fig. S9).

**Immunogenicity and immune protection against SARS-CoV-2 Challenge of S-Trimer in nonhuman primates.** The immunogenicity of adjuvanted S-Trimer was further studied in nonhuman primates (rhesus macaques). Animals ($n = 6$ per group) were vaccinated intramuscularly twice (at Day 0 and 21) with AS03-adjuvanted S-Trimer, CpG 1018 plus alum-adjuvanted S-Trimer, or a PBS vehicle control. The animals were then challenged on Day 35 with $2.6 \times 10^6$ TCID50 (60% intratracheal and 40% intranasal) SARS-CoV-2 virus and then evaluated for immune protection by various parameters.

High levels of binding and neutralizing antibody titers measured by different methods, including wild-type SARS-CoV-2 virus neutralization assay, were observed in both groups receiving adjuvanted S-Trimer (Fig. 4a–d). The boost-effect of the second dose (on Day 21) was evident, with significant increases in neutralizing antibody levels observed at Day 28 and continuing to

rise through Day 35 prior to challenge. At Day 35, neutralizing antibody titers in the AS03-adjuvanted S-Trimer group were significantly higher than levels in human convalescent sera (Fig. 4). For animals in the CpG 1018 plus alum group, despite exhibiting numerically lower binding and neutralizing antibody titers than the AS03-adjuvanted S-Trimer group, levels of antibodies were still within the range of human convalescent sera (Fig. 4). Moreover, animals in the CpG 1018 plus alum group also appeared to mount a rapid lymphocyte response that remained high 7 days after viral challenge, compared to AS03 and vehicle groups (fig. S10). Interestingly, antibody titers post-viral challenge appeared to modestly decrease following challenge at Day 40 (5 days post inoculation [dpi]), suggesting that challenge with high doses of SARS-CoV-2 may have led to rapid binding of circulating anti-Spike antibodies to the virus and subsequent clearance; a similar trend was reported in convalescent humans that were re-exposed to the virus[27].

Following challenge with SARS-CoV-2, animals in the adjuvanted S-Trimer groups were protected from body weight loss, whereas animals in the vehicle control group observed rapid body weight loss of approximately 8% through 7 dpi (Fig. 5a and fig. S11), in line with other reported studies[28]. Similarly, animals receiving adjuvanted S-Trimer appeared to be protected from increases in body temperature following SARS-CoV-2 challenge (Fig. 5b). Various blood chemistry parameters also suggested that animals in the active vaccine groups may have been protected from organ and tissue damage and other adverse effects of SARS-CoV-2 infection (fig. S12), as animals in the vehicle control group

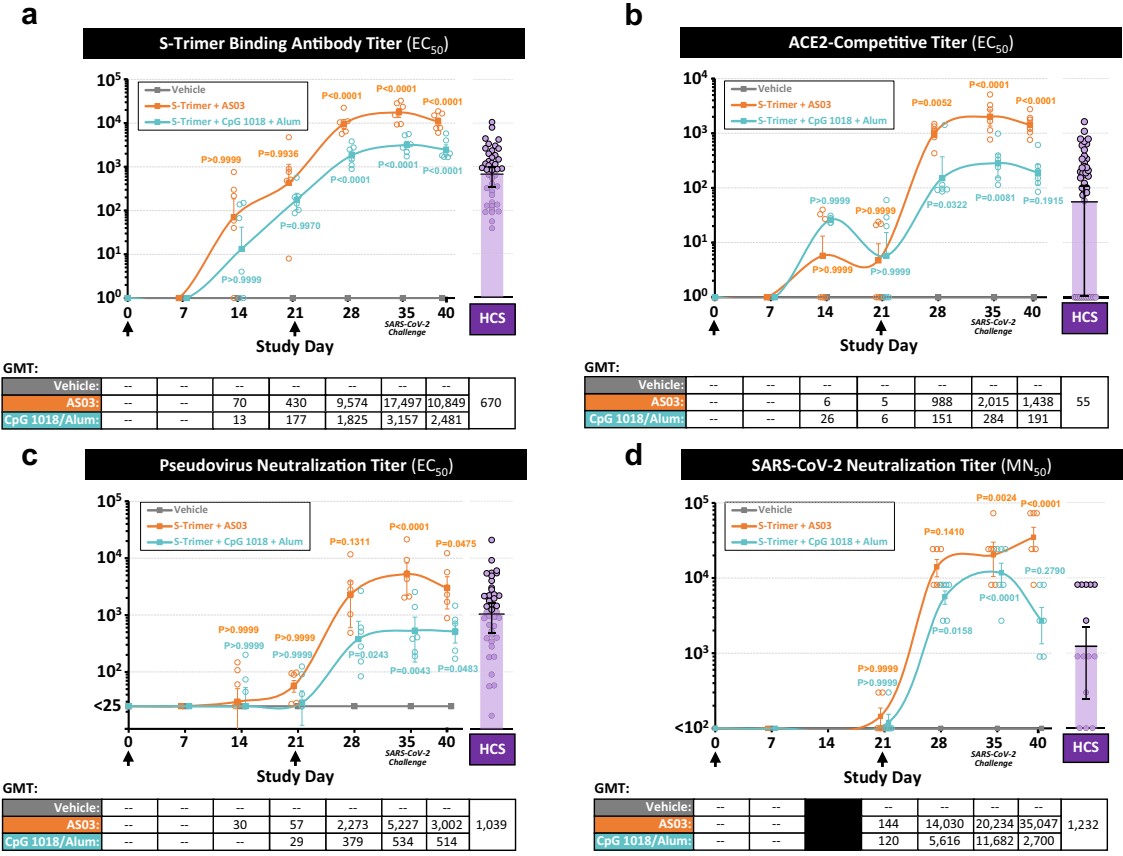

**Fig. 4 Immunogenicity of S-Trimer in nonhuman primates.** Rhesus macaques ($n = 6$/group) were immunized with 30 μg S-Trimer adjuvanted with 0.25 mL AS03, 30 μg S-Trimer adjuvanted with 1.5 mg CpG 1018 plus 0.75 mg alum, or PBS vehicle control twice on Day 0 and Day 21 and were challenged on Day 35 with $2.6 \times 10^6$ TCID50 (60% intranasal and 40% intratracheal) live SARS-CoV-2 virus. The humoral immune responses and kinetics were analyzed and compared with a human convalescent sera (HCS) panel, based on **a** S-Trimer binding antibody ELISA titers, **b** ACE2-competitive ELISA titers, **c** SARS-CoV-2 pseudovirus neutralization titers, and **d** wild-type SARS-CoV-2 virus neutralization titers. Circular points represent individual animals and humans; kinetics data are presented as geometric mean titers (GMT) ± SEM. For statistical analysis of antibody titers, all comparisons were made against the vehicle negative control group with Two-way ANOVA multiple comparisons test. $P$ values < 0.05 were considered significant.

observed increases in blood albumin (ALB), A/G ratio, AST, creatine kinase (CK), glucose (GLU), lactic acid (LAC), and triglycerides (TRIG) through 7 dpi compared to the adjuvanted S-Trimer groups.

Lung tissues were harvested at necropsy from 5 to 7 dpi and tested for SARS-CoV-2 viral loads based on genomic RNA (gRNA). Complete reduction of viral loads in lung tissues was observed in AS03 and CpG 1018 plus alum-adjuvanted S-Trimer groups, whereas viral loads were detectable in the vehicle group (Fig. 5c and fig. S13). Similar trends of reduced viral loads in animals receiving active vaccine were observed from throat swabs, anal swabs, and tracheal brushes after challenge through 7 dpi (Fig. 5d). Viral gRNA detected in nasal swabs were expected given the location of viral challenge and is not necessarily indicative of replicating virus. Histopathological analysis conducted in lung tissues and IHC staining with antibody specific to the Spike protein further confirmed the reduced SARS-CoV-2 infection in animals vaccinated with S-Trimer (Fig. 5e).

**D614G mutation in SARS-CoV-2 Spike protein does not alter receptor binding nor escape from neutralizing antibodies elicited by S-Trimer.** Since SARS-CoV-2 with D614G mutation in the Spike protein has become the predominant circulating strain in many regions of the world[29], we also produced S-Trimer with the D614G mutation. The results showed that, compared to the wild-type S-Trimer, no significant differences were observed in

ACE2 binding affinity, nor ACE2 competitive binding against anti-Spike neutralizing antibodies produced from animals immunized with wild-type S-Trimer (fig. S14).

## Discussion

**S-Trimer resembles native SARS-CoV-2 spike protein in structure and functions.** Unlike other full-length Spike proteins previously used for structural studies and vaccine development that utilized mutations introduced to abolish S1/S2 cleavage by furin protease and reportedly stabilize the protein in a prefusion form[6,30], S-Trimer is partially cleaved at the S1/S2 junction, similar to S proteins isolated from live SARS-CoV-2 virus[31] and recombinant full-length S expressed in HEK293 cells[32]. Importantly, we demonstrated that the S-Trimer vaccine candidate, with a fully wild-type S sequence from SARS-CoV-2, is not only expressed at high levels in CHO cells but also is highly glycosylated and adopts a native-like trimeric prefusion conformation[32]. N-terminal protein sequence analysis revealed that upon signal peptide removal during its biosynthesis, S-Trimer has N-terminal 14Q modified by pyroglutamate formation to protect itself from exo-protease degradation. Fusion to Trimer-Tag allows the soluble wild-type S protein to form a disulfide bond-linked homotrimer with a partially cleaved S1 that remains non-covalently bound to S-Trimer, thus preserving the crucial antigenic epitopes necessary for viral neutralization. Receptor-binding study showed that S-Trimer had high affinity to ACE2-Fc, which is one order of

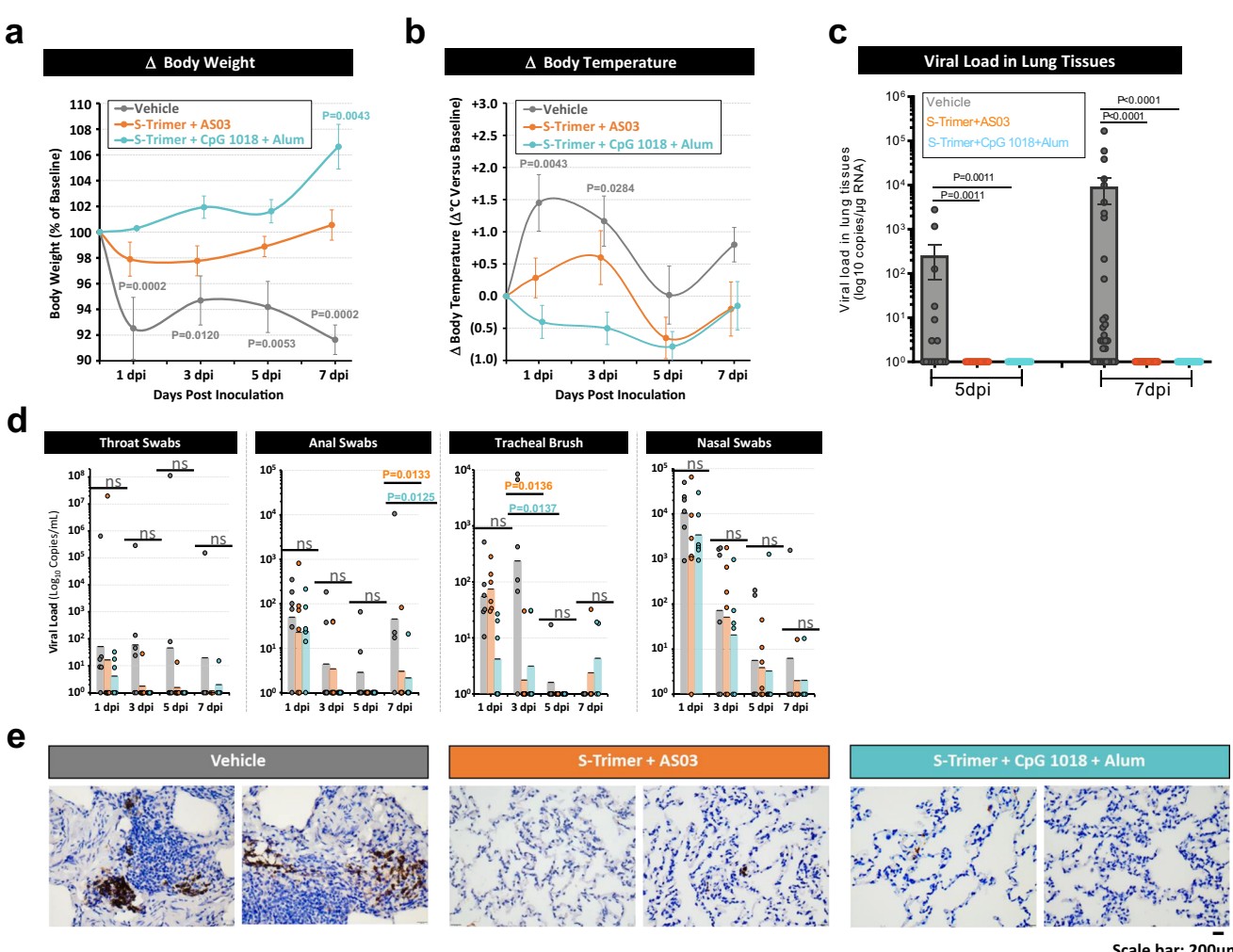

**Fig. 5 Immune protection of S-Trimer against SARS-CoV-2 challenge in nonhuman primates.** Rhesus macaques ($n = 6$/group) were immunized with 30 μg S-Trimer adjuvanted with 0.25 mL AS03, 30 μg S-Trimer adjuvanted with 1.5 mg CpG 1018 plus 0.75 mg alum, or PBS vehicle control twice on Day 0 and Day 21 and were challenged on Day 35 with $2.6 \times 10^6$ TCID50 (60% intratracheal and 40% intranasal) live SARS-CoV-2 virus. Following SARS-CoV-2 challenge, clinical observation data were collected based on **a** changes in body weight and **b** changes in body temperature at 0, 1, 3, 5 ($n = 6$) and 7 dpi ($n = 4$). **c** At necropsy at 5 dpi ($n = 2 \times 8$ samples/group) and 7 dpi ($n = 4 \times 8$ samples/group), lung tissues were collected for measurement of viral loads based on genomic RNA (gRNA). **d** Throat swab, anal swab, tracheal brush, and nasal swab specimens at 1, 3, 5 ($n = 6$) and 7 dpi ($n = 4$) were collected for measurement of viral loads based on gRNA. Body weight and body temperature data are presented as mean values ± SEM. All viral load data are presented as geometric mean values ± SEM. **e** Histopathological examinations in lungs from inoculated animals was conducted at necropsy. Lung tissues were collected and IHC staining with antibody specific to SARS-CoV-2 Spike protein was conducted. Representative specimens are shown from two independent experiments, scale bar are 200 μm. For Body weight and temperature change statistical analysis, all comparisons were compared to 0 dpi with Two-way ANOVA multiple comparison test. The statistical analysis of viral load in lung tissues with Kruskal–Wallis ANOVA with Dunn's multiple comparisons test and analysis of viral load in swabs with Two-way ANOVA multiple comparisons test were all compared to the vehicle control group. P values < 0.05 were considered significant. ns represents no significant.

magnitude higher than first reported recombinant soluble S stabilized by foldon with 2 Pro mutations[6], but similar to that reported by Walls et al.[33]. However, the differences could be attributed also to the different methods used for the affinity binding studies. The advantages of Trimer-Tag over foldon, such as fibritin from bacteria phage T4[6], for protein trimerization could be attributed to the latter being none covalently linked, nor from an abundantly secreted protein of human origin, making it less stable with also potential immunogenicity on its own when administered to human. Furthermore, unlike Trimer-Tag which can be purified via Endo180 affinity purification, there is no tailored affinity purification scheme developed for foldon which has to rely on additional tags, such as 6XHIS or Strep-Tag, to be added.

In addition to tailored-affinity purification scheme we developed for any Trimer-Tagged fusion proteins followed by further downstream purification steps typical for the production of modern biologics to ensure purity and safety (including preventative VI and VR), the current titer in bioreactor of approximately 500 mg/L would predict that billions of doses of S-Trimer antigen to be used in a COVID-19 vaccine may be produced annually from several 2000L bioreactors. The potential production output of S-Trimer antigen further supports the rationale to advance its clinical development with multiple adjuvants in parallel (such as AS03 and CpG 1018 plus alum), should supply of a single adjuvant be a limiting factor for the global supply of the vaccine (antigen plus adjuvant). A Phase I clinical trial is currently ongoing to evaluate the safety and

immunogenicity of S-Trimer with AS03 and CpG 1018 plus Alum adjuvants (NCT04405908).

**Insights from the analysis of antibody titers in human convalescent sera.** Understanding which components of the immune system are needed to confer optimal immune protection against SARS-CoV-2 infection and COVID-19 disease is critical to the development of effective vaccines. It has been reported that individuals with mild COVID-19 disease observe low or undetectable levels of neutralizing antibodies[23,24], and ~35% of SARS-CoV-2 naïve individuals have cross-reactive CD4[+] T-cell responses to SARS-CoV-2 antigens due to prior infection by other common-cold coronaviruses[34]. In this study, a clear association between higher antibody titers specific to SARS-CoV-2 and more severe disease was observed, using a panel of 41 human convalescent sera samples collected from recovered COVID-19 patients. In fact, most patients with mild COVID-19 disease did not have any detectable ACE2-competitive titers and only had low neutralizing antibody titers (Fig. 2b), implying that a strong neutralizing humoral immune response may not be the only component of the immune system that is involved in the prevention or recovery from COVID-19 disease. These observations suggest that SARS-CoV-2 could be particularly susceptible to cell-mediated immune responses, and some patients that develop rapid adaptive T-cell responses may not additionally need high levels of neutralizing antibodies to eliminate the virus. Indeed, it has been reported that patients with asymptomatic or mild COVID-19 disease develop robust T-cell immunity, while patients with severe COVID-19 disease observed T-cells at unphysiologically low levels[35] and potentially require higher levels of neutralizing antibodies in order to mount an effective recovery. While neutralizing monoclonal antibodies against SARS-CoV-2 Spike protein have been demonstrated to be protective against viral challenge in animals[36,37], it appears likely that COVID-19 vaccines inducing both humoral and cell-mediated immune responses may confer optimal protection against SARS-CoV-2.

**Adjuvanted S-Trimer induces high levels of neutralizing antibodies in rodents and protective immunity in non-human primates.** Vaccine adjuvants can contribute to achieving stronger immune responses to a viral antigen. The use of an adjuvant is of particular importance in a pandemic situation, since it could reduce the amount of antigen required per dose, allowing significantly more doses of vaccine to be produced and therefore contributing to the protection of more people. Importantly, AS03, CpG 1018 and alum adjuvants have all been utilized in commercially-licensed vaccines and have significant safety databases in clinical and post-marketing studies[38,39].

In our studies, we have observed significant adjuvant effects of AS03 and CpG 1018 plus alum, with robust high-level induction of both humoral and cell-mediated immune responses to S-Trimer in rodents and nonhuman primates. The neutralizing antibody titers from NHP against live SARS-CoV-2 after the 2nd immunization for S-Trimer with both adjuvants appeared to be similar to that of full-length S protein made from insect cells[30], but 1–2 orders of magnitude higher than those from similar studies using other vaccine approaches such as mRNA, DNA, inactivated SARS-CoV-2, RBD derived subunit and other viral vector based COVID-19 vaccine candidates[31,40–42]. It is possible that the difference could be partly due to variability in SARS-CoV-2 neutralizing antibody assays used in different vaccine studies. But the fact that we saw S-Trimer induced over 1 order of magnitude higher pseudovirus neutralizing antibody titer than that of dimeric S-Fc in a side-by-side comparison in mice and

similar high titers reported for the full-length trimeric S subunit vaccine from Novavax supports that native-like trimeric antigen conformation is crucial for COVID-19 vaccine design. This finding seems to be also consistent with an exceptional protective immunity of adjuvanted S-Trimer rendered to NHP when all COVID-19 parameters were analyzed, including disease symptoms, viral loads and biochemical parameters for tissue damages, which compared more favorably to most of the previous studies. It is interesting to note that nearly all of the previous NHP challenge studies, except AZ/Oxford, used much lower viral titers for NHP challenge studies than what we used ($2.6 \times 10^6$ virons /animal), following the first reported NHP challenge model for COVID-19[28].

Interestingly, we did observe some differences in the immune responses stimulated by these two adjuvant systems. In nonhuman primates, AS03 appeared to induce a stronger humoral immune response, inducing higher levels of neutralizing antibody titers than CpG 1018 plus alum. While antibody titers were lower in nonhuman primates for CpG 1018 plus alum (albeit still in the range of or higher than human convalescent sera), CpG 1018 plus alum did appear to potentially induce a cellular immune response (as measured by lymphocyte frequency) in nonhuman primates and was more strongly Th1-biased in rodents. However, there were no clear differences in the immune protection against SARS-CoV-2 challenge observed between the two adjuvant systems in nonhuman primates, suggesting that both adjuvants had induced sufficient and protective levels of immunity. Moreover, in another immunogenicity study in NHP, we found that S-Trimer with either AS03 or CpG 1018 alone as adjuvant could both induce Th1-biased cellular immune response, which was consistent with our data from rodent studies (fig. S15). Importantly, no signs of disease enhancement were observed, a theoretical concern for SARS-CoV-2 vaccines based on prior experience with vaccine candidates against SARS-CoV and RSV that utilized inactivated viruses[11,12]. One concern about the use of a fully human Trimer-Tag to stabilize the trimeric S antigen could be the antigenicity of the tag itself. The design of Trimer-Tag has fully taken this into consideration. Trimer-Tag comes from the C-propeptide of human type I(α) collagen which is the most abundant protein produced in the body, as such it must be under heavy negative immune selection during development. More over the C-propeptide which is responsible for initiation of collagen trimerization is proteolytically cleaved and turned over as a waste product. Consistently, data from our earlier human phase I trails for TRAIL-Trimer[18] with multiple intra cavity infusions with dosage up to 120 mg which is over 1000X that to be used for S-Trimer subunit vaccine has shown no ADA induced (anti drug antibody) in the cancer patients. Determination of antibody to Trimer-Tag has been carefully monitored in our ongoing human Phase I trials for S-Trimer with consistent result which will be reported separately in the near future.

Our data demonstrate that S-Trimer adjuvanted with either AS03 or CpG 1018 plus alum can induce robust humoral and cellular immune responses in various animal species and protective immunity against SARS-CoV-2 infection in nonhuman primates, with no signs of disease enhancement. Collectively, these results support the advancement of adjuvanted S-Trimer through human clinical studies to further demonstrate safety, immunogenicity, and vaccine efficacy. Importantly, the recombinant production of S-Trimer utilizing Trimer-Tag technology has been streamlined with the ability to be rapidly scaled-up to billions of doses annually, and the subunit vaccine can be stored at 2–8 °C (does not require frozen storage conditions). These advantages could allow S-Trimer vaccine candidate, if successful in clinical studies, to contribute significantly to the control of the COVID-19 pandemic. A Phase 1 clinical study was initiated in

June 2020 (NCT04405908), and late-stage clinical studies to evaluate vaccine efficacy and safety are also planned. Should adjuvanted S-Trimer be proven successful as a COVID-19 subunit vaccine, Trimer-Tag may become an important platform technology for rapid responses to future threats posed by any emerging enveloped RNA viruses.

## Methods

**Animal studies, facilities, and ethics statements.** Specific pathogen-free (SPF) BALB/c female mice (6–8 weeks old) for immunogenicity studies were purchased from Chengdu Dossy Experimental Animals Co., LTD and kept under standard pathogen-free conditions in the animal care center at Chengdu Hi-tech Incubation Park. All animals were allowed free access to water and diet and provided with a 12-h light/dark cycle (temperature: 16–26 °C, humidity: 40–70%). All mouse experiments were approved by the institutional animal care and use committee (IACUC) in Clover Biopharmaceuticals and were conducted according to international guidelines for animal studies. Sprague Dawley (SD) rats immunogenicity studies were performed at JOINN Laboratories Inc. (Suzhou, China), and SD rats (6–9 weeks old) were purchased from Zhejiang Vital River Laboratory Animal Technology Co., Ltd. Studies with SD rats were compliant with the policies of JOINN Laboratories Inc., the Guide for the Care and Use of Laboratory Animals (8th Edition, Institute of Laboratory Animal Resources, Commission on Life Sciences, National Research Council; National Academy Press; Washington, D.C., 2010), and the U.S. Department of Agriculture through the Animal Welfare Act (Public Law 99–198). The rhesus macaque SARS-CoV-2 challenge study was performed at Kunming Institute of Zoology (KIZ), Chinese Academy of Sciences (CAS), and male animals 3–6 years of age were used for the study. The biosafety level 3 (BSL3) lab at CAS followed the international guidelines for the animal experiment and approved by the institutional animal ethics committee of KIZ, CAS (No.: IACUC20005) prior to the studies. All study designs were reviewed by GSK.

**Human COVID-19 convalescent serum samples.** 41 human convalescent sera samples from recovered COVID-19 patients (table S1) were obtained from Public Health Clinical Center of Chengdu in Chengdu, China, under approved guidelines by the Institutional Review Board (IRB), and all patients had provided written informed consent before sera sample were collected. All convalescent sera samples were heat inactivated at 55 °C for 30 min before being used for analysis.

**Adjuvants.** AS03[38,43] was manufactured under GMP by GSK Vaccines. AS03 is an Adjuvant System containing α-tocopherol and squalene in an oil-in-water emulsion. CpG 1018[39] was manufactured under GMP by Dynavax Technologies. CpG 1018, a TLR-9 agonist, is a synthetic CpG-B class oligonucleotide having a phosphorothioate-backbone and the sequence 5′-TGACTGTGAACGTT**CG**A-GATGA-3′. Alum hydroxide was manufactured under GMP by Croda. The S-Trimer subunit vaccines were mixed with the adjuvants by gentle inversion in 1:1 ratio by volume preceding each immunization.

**Protein expression and purification.** To produce the wild-type secreted S-Trimer fusion protein, a cDNA encoding the ectodomain of wild-type SARS-CoV-2 spike (S) protein (amino acid residues 1 to 1211) (GenBank: MN908947.3) was gene-synthesized using Cricetulus griseus (Chinese hamster)-preferred codons by GenScript. The cDNA was subcloned into pTRIMER expression vector (GenHunter Corporation) at Hind III and Bgl II sites to allow in-frame fusion of the soluble S protein to Trimer-Tag (amino acid residue 1156-1406 from human Type I(α) collagen) as described previously[18]. The expression vector was transfected into GH-CHO (dhfr−/−) cell line (GenHunter Corporation) using FUGENE 6 (Roche) and grown in IMDM medium with 10% FBS. After stepwise gene amplification with increasing concentrations (0.0–10 nM) of MTX (Sigma), the clones producing the highest S-Trimer titer were then adapted to SFM-4CHO serum-free medium (GE BioSciences). The secreted S-Trimer fusion protein was then produced in 15 L bioreactors (Applikon) using a fed-batch process with Cell Boost 2 supplement (GE Hyclone) per instructions from the manufacturer. S-Trimer (D614G mut) was generated by site-directed mutagenesis using mutagenesis primer pair (Table S3) and wild-type S-Trimer expression vector as a template following the protocol of QuikChange kit (Strategene). The expression vectors were transient transfected into HEK-293F cell lines (Clover Biopharma) using PEI (Polyscience) and grown in OPM-293 CD05 medium (OPM) with OPM-293 proFeed supplement (OPM).

After harvesting the clarified cell culture medium via depth-filtration (Millipore) to remove cells, S-Trimer was purified to homogeneity by consecutive chromatographic steps. A Protein A affinity column using MabSelect PrismA (GE Healthcare) preloaded with Endo180-Fc was used to affinity-capture S-Trimer, based on the high affinity binding between Endo180 and Trimer-Tag[19] (fig. S1). After washing off unbound contaminating proteins, S-Trimer was eluted using 0.5 M NaCl in phosphate buffered saline, conditions that do not elute Endo180-Fc from Protein A. After one hour of low pH (pH 3.5) viral inactivation (VI) using acetic acid, the pH was adjusted to neutral range before loading onto Capto QXP resins (GE BioSceinces) in a flow-through mode to remove host cell DNA and residual host cell proteins (HCP). Then, a preventative viral removal (VR) step

using nanofiltration followed by a final UF/DF (Millipore) for buffer change were used to achieve the S-Trimer active drug substance (DS).

Endo180-Fc expression vector was generated by subcloning a PCR amplified cDNA encoding soluble human Endo180 (amino acid residue 1–1394)[19] into the HindIII site of pGH-hFc expression vector (GenHunter Corporation) to allow in-frame fusion to human IgG Fc. The expression vector was transfected into GH-CHO (dhfr−/−) cell line (GenHunter Corporation) using FUGENE 6 (Roche) and grown in IMDM medium with 10% FBS. After stepwise gene amplification with increasing concentrations (0.0–10 nM) of MTX (Sigma), a high titer clone was then adapted to SFM-4CHO serum-free medium (GE BioSciences). The secreted End0180-Fc fusion protein was then produced in 15 L bioreactors (Applikon) using a fed-batch process with Cell Boost 2 supplement (GE Hyclone) per instructions from the manufacturer. Endo180-Fc was purified to homogeneity by protein A affinity chromatography using MabSelect PrismA (GE Healthcare) followed by Capto QXP resins (GE BioSceinces) in a flow-through mode to remove any host cell DNA and residual host cell proteins (HCP).

ACE2-Fc and S-Fc expression vector were generated by subcloning a gene-synthesized cDNA template (GenScript) encoding soluble human ACE2 (amino acid residue 1-738, accession number: NM_001371415.1) and ectodomain of S into Hind III and Bgl II sites of pGH-hFc expression vector (GenHunter Corporation) to allow in-frame fusion to human IgG Fc, respectively. The expression vector were then stably transfected into GH-CHO (dhfr -/-) cell line and high expression clones were selected and adapted to SFM-4-CHO (Hyclone) serum-free medium and ACE2-Fc was produced in 15L bioreactors, as described for Endo180-Fc above, S-Fc was produced in 1 L flask. ACE2-Fc was purified to homogeneity from the conditioned medium using PoRos XQ column (Thermo Fisher) following the manufacturer's instructions and S-Fc was purified by protein A affinity chromatography using MabSelect PrismA (GE Healthcare).

To produce the influenza hemagglutinin (HA)-Trimer antigens, cDNA encoding the ectodomains of HA (amino acid residues 1-518 of H1N1 HA, 1-513 of H3N2 HA, 1-546 of B1 HA, and 1-547 of B2 HA) (EpiFluDatabase Accession Nos. EPI516535, EPI614444, EPI540675, EPI498048, respectively) were gene-synthesized using Cricetulus griseus -preferred codons by GenScript. Then cDNAs were cloned into pTRIMER expression vector (consistent with S-Trimer construction). pTRIMER expression vectors containing HA ectodomain-encoding sequence were transfected into GH-CHO (dhfr−/−) cell lines using FUGENE 6 and grown in IMDM medium with 10% FBS. After stepwise gene amplification with increasing concentrations (0.0–0.5 μM) of MTX (Sigma), clones producing the highest HA-Trimer titers were then adapted to SFM-4-CHO, and HA-Trimers were produced in 1L shake-flasks using fed-batch processes with CellBoost 2 supplement (GE Hyclone). After completion of the upstream cell culture process, HA-Trimers were purified from cell-free culture media after centrifugation at 3000×g for 20 min, followed by using a 5 mL Blue Sepharose column (GE Healthcare, Logan, UT, USA) under a salt-gradient (0.1–0.5 M NaCl) elution. Fractions corresponding to HA-Trimers were further polished via gel filtration using Superdex 200 (GE Healthcare) according to manufacturer's instructions to change buffer and then concentrated by ultrafiltration into PBS.

**SEC-HPLC.** The purity of S-Trimer was analyzed by Size-Exclusion Chromatography (SEC-HPLC) using Agilent 1260 Infinity HPLC with an analytic TSK gel G3000 SWxL column (Tosoh). Phosphate-buffered saline (PBS) was used as the mobile phase with $OD_{280}$ nm detection over a 20-min period at a flow rate of 1 ml/min.

**Antibodies and western blots.** Antibodies used for this study were made in house including the mouse anti SARS-CoV2 S1 monoclonal antibody, mouse anti SARS-CoV2-S2 monoclonal antibody and rabbit anti-Trimer-Tag Polyclonal antibody (Clover Biopharmaceuticals). Purified S-Trimer (0.2 μg) was analyzed by western blot on a 6% SDS-PAGE gel under reducing (+β-mercaptoethanol) conditions using the above antibodies, followed by goat anti-mouse IgG-HRP (Southern Biotech, Birmingham, AL, USA) or goat anti-Rabbit IgG-HRP (Southern Biotech, Birmingham, AL, USA). All of the above antibodies were used with 1:5000 dilutions. Reactive proteins were visualized with ECL kit following the manufacturer's protocol.

**Receptor-binding studies of S-Trimer to human ACE2.** The binding affinity of S-Trimer to ACE2 was assessed by Bio-Layer Interferometry measurements on ForteBio Octet QKe (Pall). ACE2-Fc (10 μg/mL) was immobilized on Protein A (ProA) biosensors (Pall). Real-time receptor-binding curves were obtained by applying the sensor in 2-fold serial dilutions of S-Trimer (22.5–36 μg/mL in PBS). Kinetic parameters ($K_{on}$ and $K_{off}$) and affinities ($K_D$) were analyzed using Octet software, version 12.0. Dissociation constants ($K_D$) were determined using steady state analysis, assuming a 1:1 binding model for a S-Trimer to ACE2-Fc.

**Negative staining electron microscopy.** S-Trimer was diluted to 25 μg/mL in PBS with pH adjusted to 5.5 with acetic acid and applied for 1 min onto the carbon-coated 400 CU mesh grid that had been glow-discharged at 12 mA for 20 s. The grids were negatively stained with 1% (w/v) uranyl formate at pH 4.0 for 1 min. The samples were collected through FEI Tecnai spirit electron microscope operating at 120 KeV.

**Immunogenicity analysis of S-Trimer in rodents**. BALB/c mice ($n = 7–8$/group, female) were used for immunogenicity studies. Mice were immunized intramuscularly (IM) with various doses of S-Trimer (3 µg, 9 µg, 30 µg/dose) that was non-adjuvanted or adjuvanted with 25 µL of AS03 (GSK), 10 µg of CpG 1018 (Dynavax), or 10 µg of CpG 1018 plus 50 µg of Alum (Croda). The total injection volume of the mixed vaccines (antigen+adjuvant) was 50 µL per IM dose. Two IM doses were administered (at Day 0 and Day 21). Animals were bled from the tail veins for humoral immune response analyses. Spleens were removed after killing at Day 35 for ELISpot assays.

SD rats ($n = 10$/group, half male and female) were immunized IM with various doses of S-Trimer (3 µg, 9 µg, and 30 µg/dose) that was non-adjuvanted or adjuvanted with 0.25 mL of AS03 (GSK), 1.5 mg of CpG 1018 (Dynavax), or 1.5 mg of CpG 1018 plus 0.375 mg of Alum (Croda). The total injection volume of mixed vaccines (antigen+adjuvant) was 0.5 mL per IM dose for animals receiving AS03- and CpG 1018-adjuvanted S-Trimer and 0.75 mL per IM dose for CpG 1018 plus alum-adjuvanted S-Trimer. Total injection volume for vehicle control was 0.5 mL per IM dose. Three IM immunizations were administered (at Day 0, Day 14, and Day 28). Animals from the negative control group were injected with PBS. Animals were bled from the jugular veins for humoral immune response analyses.

**Comparative immunogenicity analysis of S-Trimer verus S-Fc**. BALB/c mice ($n = 6$/group, female) were used for this comparative study. Mice were immunized intramuscularly (IM) with either 3 µg of S-Trimer or S-Fc adjuvanted with 25 µL of AS03 (GSK). The total injection volume of the mixed vaccines (antigen+adjuvant) was 50 µL per IM dose. Two IM doses were administered (at Day 0 and Day 21). Animals were bled from the tail veins for humoral immune response analysis.

**S-Trimer (or HA-Trimer) binding antibody ELISA assays**. S-Trimer binding antibody titers (or HA-Trimer binding antibody titers) in sera samples collected from immunized animals was determined by ELISA. 96-well plates (Corning) were coated with S-Trimer (or HA-Trimer) (1 µg/mL, 100 µL/well) at 4 °C overnight and blocked with 2% non-fat milk at 37°C for 2 h. Serial dilutions of the antisera were added to the wells. After incubating for 1 h at 37°C, the plates were washed three times with PBST (PBS containing 0.05% Tween-20), followed by incubating with species specific second antibody, including 1:5000 diluted goat anti-mouse IgG-HRP, 1:10,000 diluted goat anti-human IgG-HRP, 1:10,000 diluted goat anti-rat IgG-HRP (all from Southern Biotech) or 1:10,000 diluted rabbit anti-Monkey IgG-HRP (Solarbio) at 37 °C for 30 min. Plates were then washed three times with PBST and signals were developed using TMB substrate (Thermo Scientific). The colorimetric reaction was stopped after 5 min by adding 2 M HCl. The optical density (OD) was measured at 450 nm. Antibody titers (EC50) were defined as the reciprocal of the dilution from a sample at which 50% of the maximum absorbance was observed. A modified logit-log equation ("Fit Equation") was used to fit serum titration data for EC50 determination by using Origin 9 software.

**ACE2-competitive ELISA assay**. 96-well plates (Corning) were coated with 1 µg/mL ACE2-Fc (100 µL/well) at 4 °C overnight, blocked with 2% non-fat milk 37 °C for 2 h. After washing three times with PBST, the plates were incubated with S-Trimer (100 ng/mL) mixed with serially diluted antisera for 1 h at 37°C. After washing three times with PBST, the plates were incubated with 1:5000 dilution of rabbit anti-Trimer-Tag antibody (Clover Biopharmaceuticals) at 37 °C for 1 h, followed by washing three times with PBST and then a 1:20,000 dilution of goat anti-rabbit IgG-HRP (Southern Biotech). After washing three times with PBST, TMB (Thermo Scientific) was added for signal development. The percentage of inhibition was calculated as follows: % inhibition = [(A-Blank)-(P-Blank)]/(A-Blank)×100, where A is the maximum OD signal of S-Trimer binding to ACE2-Fc when no serum was present, and P is the OD signal of S-Trimer binding to ACE2-Fc in presence of serum at a given dilution. The $IC_{50}$ of a given serum sample was defined as the reciprocal of the dilution where the sample shows 50% competition.

**Pseudovirus neutralization assay**. SARS-CoV-2 pseudovirus neutralization assay was conducted as previously described[44], with some modifications. To evaluate the SARS-CoV-2 pseudovirus neutralization activity of antisera, samples were first heat inactivated for 30 min and serially diluted (3-fold), incubated with an equal volume of 325 $TCID_{50}$ pseudovirus at 37 °C for 1 h, along with virus-alone (positive control) and cell-alone (negative control). Then, freshly trypsinized Vero-E6 cells were added to each well at 20,000 cells/well. Following 24 h incubation at 37 °C in a 5% CO2 incubator, the cells were lysed and luciferase activity was determined by a Luciferase Assay System (Beyotime), according to the manufacturer's protocol. The $EC_{50}$ neutralizing antibody titer of a given serum sample was defined as the reciprocal of the dilution where the sample showed the relative light units (RLUs) were reduced by 50% compared to virus alone control wells.

**Splenocyte stimulation and ELISpot assay**. To detect antigen-specific T-cell responses, ELISpot kits (Mabtech) measuring Th1 cytokines (IFN-γ, IL-2) and Th2 cytokines (IL-4 and IL-5) were used per manufacturer's instructions. Splenocytes from immunized mice or PBMC from immunized rhesus were harvested 2 weeks after the second IM immunization. In all, $5 \times 10^5$ splenocytes or $2.5 \times 10^5$ PBMC/

well (96-well plate) were stimulated in vitro with 100 nM S-Trimer antigen. Phorbol 12-myristate 13-acetate (PMA) and ionomycin as the non-specific stimulus were added to the positive control wells, whereas the negative control well received no stimuli. After 48 h incubation, biotinylated detection antibodies from the ELISpot kits and SA-ALP/SA-HRP were added. Blots were developed by the addition of BCIP/NBT or AEC substrate solution, which produced colored spots after 5–30 min incubation in the dark. Finally, the IFN-γ, IL-2, IL-4, and IL-5 spot-forming cells (SFCs) were counted using an automatic ELISpot reader.

**SARS-CoV-2 viral challenge study in rhesus macaques**. The animal model for SARS-CoV-2 challenge in rhesus macaques was conducted as previously described[45,46]. Rhesus macaques (3–6 years old) were randomized into 3 groups of 6 animals, and immunized intramuscularly with either PBS as a negative control, or 30 µg S-Trimer adjuvanted with 0.25 mL AS03, or 30 µg S-Trimer adjuvanted with 1.5 mg CpG 1018 plus 0.75 mg alum. The total injection volume of mixed vaccines (antigen+adjuvant) was 0.5 mL per IM dose for animals receiving AS03- and CpG 1018-adjuvanted S-Trimer and 0.75 mL per IM dose for CpG 1018 plus alum-adjuvanted S-Trimer. Total injection volume for vehicle control was 0.5 mL per IM dose. All groups of animals were immunized twice (on Day 0 and 21) before viral challenge. The immunized animals were challenged with a dose of $2.6 \times 10^6$ TCID50 SARS-CoV-2 virus both intratracheally (60% of challenge dose) and intranasally (40% of challenge dose) on Day 35. SARS-CoV-2 strain 107 used was obtained from the Guangdong Provincial CDC, Guangdong, China. Blood samples were collected on Day -1, 7, 14, 21, 28, and 35 (before challenge) as well as 1, 3, 5, and 7 days post-inoculation (dpi) for antibody assays, routine blood tests, blood chemistry and hematology. Clinical observations were performed on Day 35 and 1, 3, 5, and 7 dpi, including body weight and body temperature. Tracheal brushing and various swabs (throat and anal) were collected 3, 5, and 7 dpi, and total RNA was extracted for viral load analysis by qRT-PCR. At 5 dpi, 2 rhesus macaques per group were euthanized, and at 7 dpi, the remaining animals were euthanized. Lung tissue samples were homogenized, and total RNA was extracted for viral load analysis by qRT-PCR. For detection of SARS-CoV-2 genomic RNA, primers and probes used in this experiment list in (Table S3), as previously described[45,46]. Pathological examination by H&E staining and IHC staining with a mouse monoclonal antibody specific for S1 of SARS-CoV-2 (Clover Biopharma) were conducted.

**Wild-type SARS-CoV-2 neutralization assay**. Wild-type SARS-CoV-2 neutralization assay was performed in the BSL-3 lab at Kunming Institute of Zoology, CAS. Vero-E6 cells ($2 \times 10^4$ per well) were seeded in a 96-well plate overnight. On a separate 96-well plate plate, heat-inactivated antisera (56 °C for 30 min) from rhesus macaques were serially diluted in cell culture medium in 3-fold dilutions starting from 1:100. The diluted sera were mixed with an equal volume of solution containing 100 TCID50 live SARS-CoV-2 virus in each well. After 1 h incubation at 37 °C in a 5% CO2 incubator, the virus-serum mixtures were transferred to the 96-well plate containing Vero-E6 cells and cultured in a 5% CO2 incubator at 37 °C for 6 days. Cytopathic effect (CPE) of each well was recorded under microscopes, and the neutralization titer (MN50) was calculated as the reciprocal of serum dilution required for 50% neutralization of viral infection.

**Histology and immunohistochemistry**. Lung tissues collected from the rhesus macaques after SARS-COV-2 viral challenge were fixed in 10% formalin and paraffin embedded. Sections (5 µm) were prepared and stained with hematoxylin and eosin (H&E). For SARS-CoV-2 viral detection, immunohistochemical staining for SARS-CoV2-S antigen was carried out by incubating with a 1:2000 diluted mouse monoclonal antibody specific to S1 of SARS-CoV-2 (Clover Biopharma) overnight at 4 °C. After washing with PBST, non-diluted HRP-conjugated secondary antibody (ZSGB Bio PV-6002) was added for 1 h at room temperature, the sections were developed with DAB (ZSGB Bio ZLI-9017), and mounted with Neutral Balsam for analysis under an upright microscope (BX53, Olympus).

**Statistical analysis**. Data arrangement was performed by Excel and statistical analyses were performed using the Prism 8.0 (GraphPad Software). Two-tailed Mann–Whitney tests were used to compare two experiment groups. Comparisons among multiple groups were performed using Kruskal–Wallis ANOVA with Dunn's multiple comparisons tests. Two-way ANOVA test was applied for multiple groups at different time points. $P$ values < 0.05 were considered significant. ns, no significance.

**Reporting summary**. Further information on research design is available in the Nature Research Reporting Summary linked to this article.

## Data availability

No genomic or microarray data sets were generated in this study. Reference SARS-CoV-2 *Spike* protein and *ACE2* sequences were downloaded from the NCBI database (GenBank: MN908947.3 and NM_001371415.1), *HA* sequences were downloaded from EpiFluDatabase (Accession code are: EPI516535, EPI614444, EPI540675, EPI498048). Source data are provided with this paper.

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

## Acknowledgements

The authors would like to acknowledge and thank GSK and Dynavax Technologies Corporation for providing AS03 and CpG 1018 adjuvants, respectively, for this study. We thank Lihong Chen and Xiaojun Huang at Institute of Biophysics, Chinese Academy of Sciences for assistance in conducting the negative EM studies. We would also like to thank Xiaodong Wang for helpful discussions for this study and critical review and comments for this manuscript. This work was supported by grants from Coalition for Epidemic Preparedness Innovations (CEPI).

## Author contributions

J.G.L and P.L. conceived this project and designed the overall study. D.S. oversaw all mouse studies and developed in vitro antibody/neutralizing antibody assays. X.L performed expression vector construction experiments. J.W. performed cell line transfections, clone selections, and cell-line banking experiments. X.Z. directed serum-free Fed-batch cell culture in bioreactors. P. Luo and X.Y. conducted protein purification experiments. R.X. directed animal immunogenicity studies and data analyses for SD rats. S.L. directed quality control experiments. Y.L. and X.H. performed binding affinity experiments. Q.W. performed mouse studies. J.H. generated monoclonal antibodies to S protein and, together with X. L., performed binding antibody/neutralizing antibody assays. Q.X. oversaw quality assurance of the S-Trimer production process. T.Z.S., J.B.H., and Y.T.Z. directed and conducted viral challenge studies in nonhuman primates. Y.Z., M.L., and D.L. collected and provided human convalescent sera for this study. A.H. and F.Y. conducted IHC studies. W.H. and C.Z. prepared and provided SARS-CoV-2 pseudovirus for this study. J.G.L. and P.L. analyzed the data and wrote the manuscript with input from all other authors.

## Competing interests

J.G.L. and P.L. have ownership interest in Clover Biopharmaceuticals. All other authors have no competing interests.
