## [Peer Review File · Nature Communications]

Reviewers' Comments:

Reviewer #1:

Remarks to the Author:

Jian et al.

S-Trimer, a COVID-19 subunit vaccine candidate, induces protective immunity in human primates.

In this manuscript the authors present the preparation and large-scale purification of the trimeric S protein of SARS-CoV-2 and test the preparation for specificity, immunogenicity, and protection from SARS-CoV-2 viral challenge. The data presented is of strong quality, with a good description of the S-protein purification protocol, and convincing evidence of the level of purification achieved. There is also data demonstrating that the protein is detected by a series of COVID-19 convalescent serum samples. They show a first estimation of the immunogenicity of the protein in mice and rats, where formulations of the trimer adjuvanted with ASO3, CpG, or CpG + alum show superiority vs the trimer in the absence of adjuvants. Immunogenicity is measured throughout the study measuring anti-S antibodies, ACE-2 competitive antibodies, and SARS-CoV-2 neutralizing antibodies using a pseudo virus-based assay. Both, ASO3- and CpG + alum-adjuvanted formulations gave the highest titers in all these immunological assays when compared to other preparations (trimer alone or adjuvanted with CpG) and these two preparations were then used for vaccination and challenge studies in NHPs. ELISPOTs for the detection on IL-2, IFN- γ , IL-4, and IL-5 were performed in mice to determine the Th1-Th2 balance of the response to the vaccines. NHPs vaccinated and boosted with the two immunologically strongest vaccine formulations show also a good production of anti-SARS-CoV-2 antibodies and various levels of protection from SARS-CoV-2 challenge, as indicated by the PCR assay for the detection of virus. Although the manuscript showed some strengths of these vaccine formulations for achieving high standards of immunogenicity, safety, and production, there are some important missing data that will enhance their potential and increase the quality for publication.

Main points:

Binding affinity (KD) of trimer S for ACE-2 has been reported for other preparations of S and would be important to add a comparison of the values with respect to those of other vaccines. Western Blot of the S trimer's final preparation using commercial antibodies will be a reassuring measurement of antigenicity, and protein identity.

The author mention that the trimer is stable in liquid solutions at 2-8oC. For the purpose of storage and distribution, It will be view as an important addition to show the stability of the protein in an array of conditions (pH, temperature, etc) either by ACE-2 binding ability or by immunogenicity.

When measuring S trimer-specific CMI, controls, mock vaccinated animals should be included, and graphed as fold increase in cell numbers for each cytokine over the mock vaccinated control.

There is a strong introduction of the importance of safety and the detection of vaccine-enhanced disease for respiratory viruses. However, the fact that there is increase in Th-2 cytokine-producing cells in mice vaccinated with ASO3-adjuvanted trimer seems not discussed and could represent a red flag for safety. The manuscript indicates that the CMI does not seem to be related with concentration of S. However, statistical analysis should be presented, specially for ASO3-adjuvanted preparation, since for other viruses, dose is important. In addition, a full analysis of the lung pathology post-challenge in NHPs vaccinated vs controls should be performed, with scoring representative pathological parameters and showing representative images from each group to assess rule out the presence of vaccine enhancement of disease. Protection not always correlate with decrease in pathology, as was exemplified for other respiratory viruses.

No CMI is presented for NHP to show Th1-Th2 responses. This is important from the point of view of

existent mice data.

In the challenge study shown in figure 5, there remains some vaccinated animals with virus in samples from nose, anal swab, and trachea. Does this mean that mucosal immunity induced by the vaccine is suboptimal? Also, it seems that there is a tendency of the virus to increase its presence in samples from day 5 to day 7 (nasal and tracheal brushes, and also there is some indication in the lung titration shown in figure 5C), Is there a possibility of viral rebound or second wave of replication in this model?

There is a statement on the durability of the lymphocyte response (line 193-194; and 290-291) based on the blood cell count shown on figure S8, although there is not specificity demonstrated for these cells. In addition, the term durability should be better applied at the longest extent of time when protection against or immunity to SARS-CoV-2 can be demonstrated.

Although mentioned, no statistical analysis is presented. And should be shown for figures 3, 4, 5, and suppl. Figures 6, 7, 8, 9, 10 and 11.

For clarity, figure 5C should use different bars for days 5 and 7. It is indicated that n=6, however, no enough data are shown in figure 5D.

In the discussion, the authors suggest that “.. SARS-CoV-2 could be particularly susceptible to cell-mediated responses,..”. Is any evidence that their vaccine induce that important type of response?

Reviewer #2:

Remarks to the Author:

The manuscript entitled “S-Trimer, a COVID-19 subunit vaccine candidate, induces protective immunity in nonhuman primates” by Liang GJ et.al. describes the results from preclinical studies of a subunit vaccine candidate for COVID-19. S-Trimer is a fusion antigen consists of the ecto-domain of SARS-CoV-2 Spike protein in-frame fused to Trimer-Tag (C-pro-domain of human type I(a) Collagen). S-Trimer in the presence of adjuvant induced high-levels of neutralizing antibodies as well as Th1-biased cellular immune responses in animal models from mice, rates to rhesus macaques and protected nonhuman primates from SARS-CoV-2 challenge. The approach of using a fully human Trimer-Tag with disulfide bind linkage is ingenious for stabilizing viral antigen which like most of RNA viruses is present in a trimeric confirmation. The native like antigen structure would ensure authentic immune responses to be evoked including neutralizing antibodies and cell mediated immunity by the mammals when immunized. These preclinical animal studies are quite encouraging compared to previous published studies with SARS-CoV-2 neutralizing antibody titer significantly higher than those using other vaccine strategies, including those based on mRNA, DNA and viral vectors.

This manuscript is written. The experiments and analyses are technically sound and the methods are sufficiently clear, the results are interpreted appropriately and the conclusions are supported by the data. Nevertheless, there are a few concerns that the authors should address.

1. More detail information about the Trimer-Tag, such as its source and sequence, should be provided. While the reference 18 is not the original reference about the Trimer-Tag, its original reference(s) should be cited, to make it clearer to the readers.
2. Is there any report about the use of Trimer-Tag for vaccine design and development? Can Trimer-Tag be allowed for human use by the Food & Drug Administration?
3. One of the concerns is the possible presence of autoantibodies against the Trimer-Tag itself, which could have effect on the safety of the vaccine. The authors should discuss this issue in the Discussion

section.

4. One minor suggestion is to use antibodies specific for Spike protein and Trimer-Tag, respectively, to identify S-Trimer as a fusion antigen. This could be done using either ELISA or Western blot analysis.

5. In lines 61 and 67, "VAERD" was repeatedly clarified, but in different way. Actually, another term VADE (vaccine-associated disease enhancement) was recently reported (Su S, et al. Learning from the past: development of safe and effective COVID-19 vaccines. Nat Rev Microbiol. 2020 Oct 16:1-9). VADE is more general than VAERD, to describe vaccine-associated disease enhancement. This issue should be discussed.

Reviewer #3:

Remarks to the Author:

Liang et al Report a new vaccine antigen designed by fusing "Trimer" tag to the C-terminus of SARS-CoV-2 S protein. By developing a stable cell line and a unique purification strategy, large quantity of antigen can be produced and purified. Combining with AS03 or CpG 1018 adjuvants, the antigen induced high-level neutralizing antibodies and Th1-biased immune response.

There are several concerns.

Major:

1. The major novelty of this antigen is to use "Trimer" tag. But the author didn't show the benefits of using this tag. There are no data to show whether it's better or not on stability or antigenicity, compared with other antigens (for example, RBD antigen, foldon-tagged antigen, full-length S antigen, or proline stabilized antigen).

2. The author claimed that this tag maintained the "native" conformation. Please define the meaning of "native". "Native" spike is highly unstable, easy to change into postfusion conformation.

3. Line 102 and 103. The authors claimed that the antigen is stable at 2-8 °C. Also in discussion the authors said the antigen can be stored at 2-8 °C. Please include the stability data.

Minor:

1. Line 25. "High-levels" should be "high-level".

2. Line 59-71. The difficulties of HIV vaccine should not represent the overall challenges of "RNA viruses". Polio virus and measles virus are both RNA viruses with successful vaccines. And Influenza HA head domain and coronavirus S RBD are both highly immunogenic.

3. Line 149-153. There are No detectable ACE2-competitive titers doesn't mean there are no antibodies targeting RBD. A lot of RBD-derived antibodies (such as S309) would not compete with ACE2 but are still neutralizing.

4. line 236-239. N-terminal 14Q modification would prevent the degradation, but doesn't mean S protein "can be very stable in vivo".

5. Line 239-242. "Fusion to Trimer-Tag allows the soluble wild-type S protein to form a disulfide bond-linked homotrimer with a partially-cleaved S1 that remains noncovalently bound to S-Trimer and also to maintain high affinity binding to the ACE2 receptor, thus preserving the crucial antigenic epitopes necessary for viral neutralization." This sentence is very confusing.

REVIEWER COMMENTS

Reviewer #1 (Remarks to the Author):

S-Trimer, a COVID-19 subunit vaccine candidate, induces protective immunity in human primates.

In this manuscript the authors present the preparation and large-scale purification of the trimeric S protein of SARS-CoV-2 and test the preparation for specificity, immunogenicity, and protection from SARS-CoV-2 viral challenge. The data presented is of strong quality, with a good description of the S-protein purification protocol, and convincing evidence of the level of purification achieved. There is also data demonstrating that the protein is detected by a series of COVID-19 convalescent serum samples. They show a first estimation of the immunogenicity of the protein in mice and rats, where formulations of the trimer adjuvanted with ASO3, CpG, or CpG + alum show superiority vs the trimer in the absence of adjuvants. Immunogenicity is measured throughout the study measuring anti-S antibodies, ACE-2 competitive antibodies, and SARS-CoV-2 neutralizing antibodies using a pseudo virus-based assay. Both, ASO3- and CpG + alum-adjuvanted formulations gave the highest titers in all these immunological assays when compared to other preparations (trimer alone or adjuvanted with CpG) and these two preparations were then used for vaccination and challenge studies in NHPs. ELISPOTs for the detection on IL-2, IFN-gamma, IL-4, and IL-5 were performed in mice to determine the Th1-Th2 balance of the response to the vaccines. NHPs vaccinated and boosted with the two immunologically strongest vaccine formulations show also a good production of anti-SARS-CoV-2 antibodies and various levels of protection from SARS-CoV-2 challenge, as indicated by the PCR assay for the detection of virus. Although the manuscript showed some strengths of these vaccine formulations for achieving high standards of immunogenicity, safety, and production, there are some important missing data that will enhance their potential and increase the quality for publication.

We very appreciate the referee's overall positive assessment of our work and constructive suggestions. We have now revised the manuscript accordingly as below?

Main points:

Binding affinity (KD) of trimer S for ACE-2 has been reported for other preparations of S and would be important to add a comparison of the values with respect to those of other vaccines. Western Blot of the S trimer's final preparation using commercial antibodies will be a reassuring measurement of antigenicity, and protein identity.

1. Regarding comparison of our ACE2 receptor binding affinity to other published prep, we now have added it in the discussion as suggested.
2. Western blot identification of S-Trimer with antibodies specific to S1, S2 and Trimer-Tag, respectively, have been included in **Fig. S1D**.

The author mention that the trimer is stable in liquid solutions at 2-8oC. For the purpose of storage and distribution, It will be view as an important addition to show the stability of the

protein in an array of conditions (pH, temperature, etc) either by ACE-2 binding ability or by immunogenicity.

The stability data of our highly purified S-Trimer antigen has been added as **Fig. S2**. Antigen stability is a critical quality attribute (CQ) for any commercial drug or vaccine product and we have studied it under various conditions. We have now accumulated long term stability data after 6 months which showed excellent stability profile at 2-8 °C, and S-Trimer is also stable for at least 6 weeks at elevated temperatures as high as 40 °C. Longer term stability analysis is still on going.

When measuring S trimer-specific CMI, controls, mock vaccinated animals should be included, and graphed as fold increase in cell numbers for each cytokine over the mock vaccinated control.

All CMI data from vaccinated animals have been subtracted by the negative control background (no cytokine stimulation) which was essential near 0 (no to few stainings) as cells from non – vaccinated animals in method validation phase of the study. This was clarified in the Fig legends. The CMI data shown is clearly vaccination dependent.

There is a strong introduction of the importance of safety and the detection of vaccine-enhanced disease for respiratory viruses. However, the fact that there is increase in Th-2 cytokine-producing cells in mice vaccinated with ASO3-adjuvanted trimer seems not discussed and could represent a red flag for safety.

It is known that GSK ASO3 adjuvant induced both Th1 and Th2 responses during its 2009 pandemic flu vaccine studies and the vaccine was approved for clinical use based on good safety profile. Here we showed that S-Trimer+ASO3 induced balanced Th1 and Th2 responses, instead of Th2 biased phenomenon which could be a concern. Our result is consistent with previous GSK studies on ASO3 adjuvant.

The manuscript indicates that the CMI does not seem to be related with concentration of S. However, statistical analysis should be presented, specially for ASO3-adjuvanted preparation, since for other viruses, dose is important.

We have conducted statistical analysis which showed no statistical difference existed among 3, 9, and 30 ug antigen plus ASO3. It is possible further decrease in the amount of antigen would lead to assess the linear range of response in CMI, which has been clarified in the text. We will evaluate this in future studies.

In addition, a full analysis of the lung pathology post-challenge in NHPs vaccinated vs controls should be performed, with scoring representative pathological parameters and showing representative images from each group to assess rule out the presence of vaccine enhancement of disease. Protection not always correlate with decrease in pathology, as was exemplified for other respiratory viruses.

We have reviewed all tissue sections from the lung pathology post-challenge in NHP vaccinated vs. vehicle controls, and found no statistical difference in lung abnormality as pointed out by the

referee, possibly due to very mild symptoms evoked by SARS-Cov-2 in this animal model. Since there were no normal control animals included that were not challenged by the virus, it is impossible to differentiate any minor apparent morphological changes among different groups. However, we did show with S1 viral specific mAb, more clustered staining was mainly found in vehicle control group, and representative results were showed in Fig.5. Since tissue sections comes from a very small region of the lung of each animal procured in a P3 lab, any further statistical analysis will be unlikely reflecting the actual outcome. The fact that all parameters including phenotypical symptoms, biochemical serum markers as well as viral load data all suggest that ADE is not involved in animals vaccinated.

No CMI is presented for NHP to show Th1-Th2 responses. This is important from the point of view of existent mice data.

Since we have extensive CMI data from rodent studies, and a separate NHP study prior to the challenge study, we did not include the CMI analysis in the P3 lab. We have now included CMI data from NHP for the initial NHP immunogenicity study in **Fig. S15**. The result showed both AS03 and CpG adjuvanted S-Trimer induced Th1 biased CMI.

In the challenge study shown in figure 5, there remains some vaccinated animals with virus in samples from nose, anal swab, and trachea. Does this mean that mucosal immunity induced by the vaccine is suboptimal? Also, it seems that there is a tendency of the virus to increase its presence in samples from day 5 to day 7 (nasal and tracheal brushes, and also there is some indication in the lung titration shown in figure 5C), Is there a possibility of viral rebound or second wave of replication in this model?

Since the P3 lab where the assays were conducted did not establish a limit of detection (LOD) for the viral load assay, it is possible the slightly increased viral load in tracheal brush and anal swabs following the challenge could be due to errors in the assays. No such irregularity was observed in throat nor nasal swabs.

There is a statement on the durability of the lymphocyte response (line 193-194; and 290-291) based on the blood cell count shown on figure S8, although there is not specificity demonstrated for these cells. In addition, the term durability should be better applied at the longest extent of time when protection against or immunity to SARS-CoV-2 can be demonstrated.

We have removed “durable response” as suggested

Although mentioned, no statistical analysis is presented. And should be shown for figures 3, 4, 5, and supl. Figures 6, 7, 8, 9, 10 and 11.

We have now included statistical analysis for the Figs indicated above.

For clarity, figure 5C should use different bars for days 5 and 7. It is indicated that n=6, however, no enough data are shown if figure 5D.

We have revised Figure 5C. On 0-5 dpi, there were 6 animals/group, but only 4 animals remained per group on 7 dpi based on the study design (2 animals were sacrificed on 5 dpi for analysis).

In the discussion, the authors suggest that “.. SARS-CoV-2 could be particularly susceptible to cell-mediated responses,..”. Is any evidence that their vaccine induce that important type of response?

We have removed this speculation, since neutralizing antibody titers for the CPG+Alum group could be high enough to render protection from the viral infection.

Reviewer #2 (Remarks to the Author):

The manuscript entitled “S-Trimer, a COVID-19 subunit vaccine candidate, induces protective immunity in nonhuman primates” by Liang GJ et.al. describes the results from preclinical studies of a subunit vaccine candidate for COVID-19. S-Trimer is a fusion antigen consists of the ecto-domain of SARS-CoV-2 Spike protein in-frame fused to Trimer-Tag (C-pro-domain of human type I(a) Collagen). S-Trimer in the presence of adjuvant induced high-levels of neutralizing antibodies as well as Th1- biased cellular immune responses in animal models from mice, rates to rhesus macaques and protected nonhuman primates from SARS-CoV-2 challenge. The approach of using a fully human Trimer-Tag with disulfide bind linkage is ingenious for stabilizing viral antigen which like most of RNA viruses is present in a trimeric confirmation. The native like antigen structure would ensure authentic immune responses to be evoked including neutralizing antibodies and cell mediated immunity by the mammals when immunized. These preclinical animal studies are quite encouraging compared to previous published studies with SARS-CoV-2 neutralizing antibody titer significantly higher than those using other vaccine strategies, including those based on mRNA, DNA and viral vectors.

This manuscript is well written. The experiments and analyses are technically sound and the methods are sufficiently clear, the results are interpreted appropriately and the conclusions are supported by the data. Nevertheless, there are a few concerns that the authors should address.

1. More detail information about the Trimer-Tag, such as its source and sequence, should be provided. While the reference 18 is not the original reference about the Trimer-Tag, its original reference(s) should be cited, to make it clearer to the readers.

The nature of Trimer-Tag (amino acid residue 1156-1406 from human Type I(a) collagen) was further clarified in the Method section.

2. Is there any report about the use of Trimer-Tag for vaccine design and development? Can Trimer-Tag be allowed for human use by the Food & Drug Administration?

Trimer-Tag from fully human Type I collagen has been used before for creating TRAIL-Trimer (ref 18) which has been approved for multiple phase I clinical trials in Australia and China to treat malignant ascites and pleural effusion with excellent safety profile.

Furthermore, S-Trimer described in this work has also entered Phase I clinical trials on June 19, 2020 in Australia and 150 healthy volunteers have been vaccinated twice with the best safety profile and best in class immunity (neutralizing antibody titer and CMI) reported so far and the results have been submitted for publication and posted on MedRxiv (<https://www.medrxiv.org/content/10.1101/2020.12.03.20243709v1>). As a result, CEPI has pledged on Nov. 3, 2020 for a total of \$328 million to support our global Phase 2/3 trials (https://cepi.net/news_cepi/cepi-extends-partnership-with-clover-to-fund-covid-19-vaccine-candidate-through-global-phase-2-3-study-to-licensure/).

3. One of the concerns is the possible presence of autoantibodies against the Trimer-Tag itself, which could have effect on the safety of the vaccine. The authors should discuss this issue in the Discussion section.

We are keenly aware of this potential risk. We have discussed this concern in the discussion as suggested.

4. One minor suggestion is to use antibodies specific for Spike protein and Trimer-Tag, respectively, to identify S-Trimer as a fusion antigen. This could be done using either ELISA or Western blot analysis.

As indicated in response to Referee No. 1, Western blot identification of S-Trimer with antibodies specific to S1, S2 and Trimer-Tag, respectively, have been included in **Fig. S1D**.

5. In lines 61 and 67, "VAERD" was repeatedly clarified, but in different way. Actually, another term VADE (vaccine-associated disease enhancement) was recently reported (Su S, et al. Learning from the past: development of safe and effective COVID-19 vaccines. Nat Rev Microbiol. 2020 Oct 16:1-9). VADE is more general than VAERD, to describe vaccine-associated disease enhancement. This issue should be discussed.

While there may be multiple terms and acronyms used to refer to vaccine-associated disease enhancement in the context of COVID-19, regulatory agencies including U.S. FDA ('Development and Licensure of Vaccines to Prevent COVID-19: Guidance for Industry', issued on June 2020) uses 'Vaccine-Associated Enhanced Respiratory Disease' (VAERD) to describe the main theoretical risk believed to be relevant for COVID-19 vaccines, based on observations of prior vaccines for other coronaviruses and RSV. We agree that VADE is a broader term which may encompass other phenomenon such as ADE, but in the manuscript we focus on 'VAERD' due to current guidance from regulatory agencies and learnings from prior coronavirus vaccines.

Reviewer #3 (Remarks to the Author):

Liang et al Report a new vaccine antigen designed by fusing "Trimer" tag to the C-terminus of SARS-CoV-2 S protein. By developing a stable cell line and a unique purification strategy, large quantity of antigen can be produced and purified. Combining with AS03 or CpG 1018 adjuvants, the antigen induced high-level neutralizing antibodies and Th1-biased immune response. There are several concerns.

Major:

1. The major novelty of this antigen is to use “Trimer” tag. But the author didn’t show the benefits of using this tag. There are no data to show whether it’s better or not on stability or antigenicity, compared with other antigens (for example, RBD antigen, foldon-tagged antigen, full-length S antigen, or proline stabilized antigen).

The essence of Trimer-Tag is to stabilize the ecto-domain of the Spike protein of SARS-Cov-2 in its native trimeric form so the antigen could evoke a better immune response when used as a vaccine. To address the referee’s concern, we have included data (**Fig. S8**) on immunogenicity in mice using the spike protein (ecto-domain) fused to human Ig-G1 Fc (S-Fc) as a control, which showed the dimeric S antigen is significantly inferior to S-Trimer in eliciting neutralizing antibody. The result is consistent with data previously reported to the RBD based vaccine candidates.

2. The author claimed that this tag maintained the “native” conformation. Please define the meaning of “native”. “Native” spike is highly unstable, easy to change into postfusion conformation.

We have clearly stated in the discussion that the native conformation being the **prefusion** form of the Wild-type **trimeric** spike protein. (ref 32). Moreover, the Cyro EM structure of S-Trimer has been submitted for publication (ref. 22) and the structure is near identical to the Cryo EM structure of the prefusion form of the the full length Spike protein published in Science from Bing Chen lab at Harvard Med school (ref 32). We have further clarified this in the introduction.

3. Line 102 and 103. The authors claimed that the antigen is stable at 2-8 °C. Also in discussion the authors said the antigen can be stored at 2-8 °C. Please include the stability data.

As replied to Referee No. 1, the stability data is included as Fig. S2.

Minor:

1. Line 25. “High-levels” should be “high-level”.

Corrected as suggested

2. Line 59-71. The difficulties of HIV vaccine should not represent the overall challenges of “RNA viruses”. Polio virus and measles virus are both RNA viruses with successful vaccines. And Influenza HA head domain and coronavirus S RBD are both highly immunogenic.

We referred enveloped RNA viruses, and Polio virus is non-enveloped. Measles and Influenza both have high density of HA on their envelopes, which may explain why attenuated or inactivated viruses have been successful, while HIV and coronaviruses have a few to fewer than 30 antigen spikes. Also RBD that are expressed not in trimeric form are much less immunogenic. Nonetheless, we have clarified in the discussion that this is attributed to some enveloped RNA viruses.

3. Line 149-153. There are No detectable ACE2-competitive titers doesn't mean there are no antibodies targeting RBD. A lot of RBD-derived antibodies (such as S309) would not compete with ACE2 but are still neutralizing.

We did not state that low ACE2-competing antibody titer meant no antibody targeting RBD.

4. line 236-239. N-terminal 14Q modification would prevent the degradation, but doesn't mean S protein "can be very stable in vivo".

We have removed "*in vivo*". We are the first to discover this modification and indeed during two weeks in high density serum-free cell culture in a bioreactor, the S-Trimer is highly stable with intact modified 14Q as N-terminus.

5. Line 239-242. "Fusion to Trimer-Tag allows the soluble wild-type S protein to form a disulfide bond-linked homotrimer with a partially-cleaved S1 that remains noncovalently bound to S-Trimer and also to maintain high affinity binding to the ACE2 receptor, thus preserving the crucial antigenic epitopes necessary for viral neutralization." This sentence is very confusing.

We have shortened the running sentence as suggested:

"Fusion to Trimer-Tag allows the soluble wild-type S protein to form a disulfide bond-linked homotrimer with a partially-cleaved S1 that remains noncovalently bound to S-Trimer, thus preserving the crucial antigenic epitopes necessary for viral neutralization."

Reviewers' Comments:

Reviewer #1:

Remarks to the Author:

Jian et al.

S-Trimer, a COVID-19 subunit vaccine candidate, induces protective immunity in human primates.

This corrected version of the manuscript has addressed several concerns. However, some points remain to be clarified.

It is not clear what is compared in figures 3 and S15. It seems that the comparison is between different cytokines within the same vaccination group. These figures should statistically compare each cytokine expression between different groups to determine if a vaccine increase the chance of Th1 or Th2 vias, respectively.

It would be appreciated by the reader if the definition of the p values is included in each figure legend that required it.

Line 107's statement "...at least 6 months depending of the purity". This is not shown if FS2, which is not comparing different purity preparations of the S protein.

Few misspelled words detected

FS2 legend: antigen

Line 349: comparison

Line 356: should be more appropriate 2.6 x 10⁶ instead of 2.6 Millions. In addition, it is not clear that the work virions apply here.

Line 750: including.

Reviewer #2:

Remarks to the Author:

The authors have satisfactorily addressed all my concerns and revised the manuscript accordingly. Therefore, I recommend publication of this manuscript.

Reviewer #3:

Remarks to the Author:

Please see attached.

REVIEWER COMMENTS

Reviewer #1 (Remarks to the Author):

Jian et al.

S-Trimer, a COVID-19 subunit vaccine candidate, induces protective immunity in human primates.

This corrected version of the manuscript has addressed several concerns. However, some points remain to be clarified.

It is not clear what is compared in figures 3 and S15. It seems that the comparison is between different cytokines within the same vaccination group. These figures should statistically compare each cytokine expression between different groups to determine if a vaccine increase the chance of Th1 or Th2 vias, respectively.

We have revised the figure 3 and figS15 following the reviewer's suggestion. In fig3 D, we have conducted the statistic analysis between the vaccination group and non-vaccinated group for all cytokines. In fig S15, the comparisons were compared to groups vaccinated without adjuvant. □

It would be appreciated by the reader if the definition of the p values is included in each figure legend that required it.

We have added the definition of p value in each figure legends where it applies.

Line 107's statement "...at least 6 months depending of the purity". This is not shown if FS2, which is not comparing different purity preparations of the S protein.

We have revised the figS2. As suggested.

Few misspelled words detected

FS2 legend: antigen

Line 349: comparison

Line 356: should be more appropriate 2.6 x 10⁶ instead of 2.6 Millions. In addition, it is not clear that the work virions apply here.

Line 750: including.

All of the above mentioned misspelled words have been corrected.

Reviewer #2 (Remarks to the Author):

The authors have satisfactorily addressed all my concerns and revised the manuscript accordingly. Therefore, I recommend publication of this manuscript.

We very much appreciate Reviewer #2 for finding the manuscript now in acceptable form.

Reviewer #3 (Remarks to the Author):

Liang et al Report a new vaccine antigen designed by fusing “Trimer” tag to the C-terminus of SARS-CoV-2 S protein. By developing a stable cell line and a unique purification strategy, large quantity of antigen can be produced and purified. Combining with AS03 or CpG 1018 adjuvants, the antigen induced high-level neutralizing antibodies and Th1-biased immune response.

There are several concerns.

Major: 1. The major novelty of this antigen is to use “Trimer” tag. But the author didn't show the

benefits of using this tag. There are no data to show whether it's better or not on stability or antigenicity, compared with other antigens (for example, RBD antigen, foldon-tagged antigen, full-length S antigen, or proline stabilized antigen).

The essence of Trimer-Tag is to stabilize the ecto-domain of the Spike protein

of SARS-Cov-2 in its native trimeric form so the antigen could evoke a better immune response when used as a vaccine. To address the referee's concern, we have included data (Fig. S8) on immunogenicity in mice using the spike protein (ecto-domain) fused to human Ig-G1 Fc (S-Fc) as a control, which showed the dimeric S antigen is significantly inferior to S-Trimer in eliciting neutralizing antibody. The result is consistent with data previously reported to the RBD based vaccine candidates.

Fc is a dimer and will definitely influence the formation of S trimer. How about foldon-tagged S, which has been used widely? are there any advances to use Trimer tag instead of foldon?

The advantage of using Trimer-Tag for stabilizing trimerization of any secreted human proteins have been discussed in our initial publication on TRAIL-Trimer (Ref 18). However, we have further stressed them again (covalently linked, fully human, from one of the most abundantly secreted proteins in the body and with tailored affinity purification scheme using Endo180 etc), in comparison to foldon in the Discussion.

2. The author claimed that this tag maintained the "native" conformation. Please define the meaning of "native". "Native" spike is highly unstable, easy to change into postfusion conformation.

We have clearly stated in the discussion that the native conformation being the prefusion form of the Wild-type trimeric spike protein. (ref 32). Moreover, the Cryo EM structure of S-Trimer has been submitted for publication (ref. 22) and the structure is near identical to the Cryo EM structure of the prefusion form of the the full length Spike protein published in Science from Bing Chen lab at Harvard Med school (ref 32). We have further clarified this in the introduction.

3. Line 102 and 103. The authors claimed that the antigen is stable at 2-8 °C. Also in discussion the authors said the antigen can be stored at 2-8 °C. Please include the stability data.

As replied to Referee No. 1, the stability data is included as Fig. S2.

Please include the SEC-HPLC raw data, and answer the questions below. □ What does this percentage mean? Does it mean the percentage of protein fraction at ~700 kd? If

so, it's unbelievable that the protein can maintain prefusion after 6 weeks' incubation at 40°C. The author should confirm the prefusion conformation of the sample using negative-stain EM.

Previous data on stability of S-Trimer were summarized from multiple lots of S-Trimer produced. The percentage represents of the ratio of S-Trimer antigen peak area (S-Trimer + cleave S1 peak)- any impurity peak area / total peak area.

Here, as suggested by the referee, we have revised Fig. S2 to include SEC-HPLC data to better illustrate the excellent stability of S-Trimer in solution. It should be noted that the intended storage temperature of S-Trimer for clinical use is 2-8 °C. instead of at 40 °C which was only intended for accelerating stability analysis. Negative EM of S-Trimer has been presented in Fig. S3. For stability studies and batch release of a vaccine antigen, it is standard to SEC-HPLC to assess the purity, size and integrity of the antigen which has been validated with immunogenicity studies (S binding antibody titer and ACE2 competition ELISA). Longer term stability analysis is on going.

Minor: □ 1. Line 25. "High-levels" should be "high-level".

Corrected as suggested

2. Line 59-71. The difficulties of HIV vaccine should not represent the overall challenges of "RNA viruses". Polio virus and measles virus are both RNA viruses with successful vaccines. And Influenza HA head domain and coronavirus S RBD are both highly immunogenic.

We referred enveloped RNA viruses, and Polio virus is non-enveloped. Measles and Influenza both have high density of HA on their envelopes, which

may explain why attenuated or inactivated viruses have been successful, while HIV and coronaviruses have a few to fewer than 30 antigen spikes. Also RBD that are expressed not in trimeric form are much less immunogenic. Nonetheless, we have clarified in the discussion that this is attributed to some enveloped RNA viruses.

3. Line 149-153. There are No detectable ACE2-competitive titers doesn't mean there are no antibodies targeting RBD. A lot of RBD-derived antibodies (such as S309) would not compete with ACE2 but are still neutralizing.

We did not state that low ACE2-competiing antibody titer meant no antibody targeting RBD.

High neutralizing titer with no ACE2 competition SUGGESTS there are no ACE2 competing antibodies. But this doesn't necessarily mean these antibodies are targeting NTD or S2. They could still target RBD without blocking ACE2 binding, such as the antibody S309 from Vir. This sentence should be modified. Otherwise, I suggest the author to deplete all RBD antibodies with RBD protein, and then test the neutralizing titer.

As suggested, we have modified the sentence as “ ..either some RBD binding antibodies that do not interfere with ACE2 receptor binding exist, or other domains such as NTD and S2 may also be important antigenic epitopes for viral neutralization..”

4. line 236-239. N-terminal 14Q modification would prevent the degradation, but doesn't mean S protein “can be very stable in vivo”.

We have removed “in vivo”. We are the first to discover this modification and indeed during two weeks in high density serum-free cell culture in a bioreactor, the S-Trime is highly stable with intact modified 14Q as N-terminus.

N-terminal 14Q modification doesn't SUGGEST S protein can be very stable. The stability of S protein is mainly determined by the prefusion to postfusion conformational change, instead of N- terminal degradation. The author should modify this sentence.

We have modified the sentence by deleting "... suggesting that S protein from SARS-CoV-2 can be very stable".

5. Line 239-242. "Fusion to Trimer-Tag allows the soluble wild-type S protein to form a disulfide bond-linked homotrimer with a partially-cleaved S1 that remains noncovalently bound to S-Trimer and also to maintain high affinity binding to the ACE2 receptor, thus preserving the

crucial antigenic epitopes necessary for viral neutralization." This sentence is very confusing.

We have shortened the running sentence as suggested:

"Fusion to Trimer-Tag allows the soluble wild-type S protein to form a disulfide bond-linked homotrimer with a partially-cleaved S1 that remains noncovalently bound to S-Trimer, thus preserving the crucial antigenic epitopes necessary for viral neutralization."

Other minor comments:

1. Line 251-254. Please confirm and modify: □ Ref6 and ref 33 were both using 2P+foldon for structural study. For affinity, ref6 was doing SPR using S-2P, while ref33 was doing BLI using RBD.

This has been clarified as suggested.

2. Line 65. Native trimeric gp120 should be gp140.
Corrected.

Reviewers' Comments:

Reviewer #1:

Remarks to the Author:

Thank you for your corrections. I do not have more suggestions.

Reviewer #3:

Remarks to the Author:

All my concerns have been addressed. Therefore, I recommend publication of this manuscript.